# High-efficiency RNA-based reprogramming of human primary fibroblasts

Igor Kogut[1,2], Sandra M. McCarthy[1,2], Maryna Pavlova[1,2], David P. Astling[3], Xiaomi Chen[1,2], Ana Jakimenko[1,2], Kenneth L. Jones[4], Andrew Getahun[5], John C. Cambier[5], Anna M.G. Pasmooij[6], Marcel F. Jonkman[6], Dennis R. Roop[1,2] & Ganna Bilousova[1,2,7]

Induced pluripotent stem cells (iPSCs) hold great promise for regenerative medicine; however, their potential clinical application is hampered by the low efficiency of somatic cell reprogramming. Here, we show that the synergistic activity of synthetic modified mRNAs encoding reprogramming factors and miRNA-367/302s delivered as mature miRNA mimics greatly enhances the reprogramming of human primary fibroblasts into iPSCs. This synergistic activity is dependent upon an optimal RNA transfection regimen and culturing conditions tailored specifically to human primary fibroblasts. As a result, we can now generate up to 4,019 iPSC colonies from only 500 starting human primary neonatal fibroblasts and reprogram up to 90.7% of individually plated cells, producing multiple sister colonies. This methodology consistently generates clinically relevant, integration-free iPSCs from a variety of human patient's fibroblasts under feeder-free conditions and can be applicable for the clinical translation of iPSCs and studying the biology of reprogramming.

[1] Department of Dermatology, University of Colorado School of Medicine, Anschutz Medical Campus, 12801 East 17th Avenue, Aurora, CO 80045, USA. [2] Charles C. Gates Center for Regenerative Medicine, University of Colorado School of Medicine, Anschutz Medical Campus, 12800 East 19th Avenue, Aurora, CO 80045, USA. [3] Department of Biochemistry and Molecular Genetics, University of Colorado School of Medicine, Anschutz Medical Campus, 12801 East 17th Avenue, Aurora, CO 80045, USA. [4] Department of Pediatrics, University of Colorado School of Medicine, Anschutz Medical Campus, 12800 East 19th Avenue, Aurora, CO 80045, USA. [5] Department of Immunology and Microbiology, University of Colorado School of Medicine, Anschutz Medical Campus, 12800 East 19th Avenue, Aurora, CO 80045, USA. [6] Department of Dermatology, University Medical Center, Groningen AB21 Hanzeplein 1, 9713 GZ Groningen, The Netherlands. [7] Linda Crnic Institute for Down Syndrome, University of Colorado School of Medicine, Anschutz Medical Campus, 12700 East 19th Avenue, Aurora, CO 80045, USA. Correspondence and requests for materials should be addressed to D.R.R. (email: dennis.roop@ucdenver.edu) or to G.B. (email: ganna.bilousova@ucdenver.edu)

Reprogramming somatic cells into induced pluripotent stem cells (iPSCs) through ectopic expression of the transcription factors OCT4, KLF4, SOX2, and cMYC (known as the Yamanaka factors) provides an unlimited supply of cells with embryonic stem cell (ESC)-like properties[1–4]. Despite great advances in developing reprogramming approaches, the efficiency of iPSC generation remains relatively low[5,6], hampering the potential application of iPSC technology in clinical and research settings.

To overcome low reprogramming efficiency, a variety of reprogramming modulators have been identified to date. However, when combined with the Yamanaka factors, many of these modulators produce only a modest enhancement of overall reprogramming efficiency[6–9], while others function exclusively on murine cells[10–12]. The expression level and stoichiometry of reprogramming factors may also influence the efficiency of reprogramming[13]; however, only a few reprogramming protocols allow for the precise control over these parameters. Reprogramming with synthetic capped mRNAs containing modified nucleobases (mod-mRNA) is the most promising among these approaches due to its relatively high efficiency (up to 4.4%)[14,15], low activation of an innate antiviral response[14], and production of high-quality, clinically relevant iPSCs[6]. Although the mod-mRNA-based approach successfully reprograms established, long-lived fibroblast cell lines such as BJs[14,15], this method is inconsistent when applied to freshly isolated patient's cells[6]. This observation suggests that the conditions optimized for established fibroblast lines may not fully support the reprogramming of primary cells due to differences in culturing conditions, RNA transfection efficiency, and gene expression profiles between these cell types[16]. Thus, an optimal regimen for the mod-mRNA-based reprogramming of human primary fibroblasts has not been established.

Here, we sought to overcome the inconsistencies of the mod-mRNA-based reprogramming approach and develop an efficient, integration-free reprogramming protocol adapted specifically to human primary fibroblasts. To accomplish this goal, we supplemented the mod-mRNA cocktail of reprogramming factors[15] with ESC-specific miRNA-367/302s[17] as mature miRNA mimics. The cocktail of mature miRNA-367/302s mimics is referred to as m-miRNAs in this study. The miRNAs-367/302s family of miRNAs has been previously shown to induce pluripotency in somatic cells[17] and enhance the efficiency of the mod-mRNA-based reprogramming[6,7]. We also optimized the RNA transfection regimen, cell seeding, and culturing conditions during reprogramming. We show that the combination of the reprogramming mod-mRNAs and m-miRNAs enhances the generation of iPSCs from human primary fibroblasts in a synergistic manner. Because of this synergism, we can reprogram human patient's fibroblasts with an efficiency that surpasses all previously published integration-free protocols. Our protocol employs feeder-free culture conditions, produces clinically relevant iPSCs, and is capable of reprogramming even an individually plated human cell. Our data suggest that the reprogramming efficiency of other cell types may be greatly improved by optimizing both culture and RNA transfection conditions.

## Results

**Optimized delivery of RNAs enhances reprogramming**. We speculated that the efficiency of mod-mRNA-based reprogramming could be improved by incorporating ESC-specific m-miR-NAs. In addition, since high cell cycling was previously shown to promote more efficient reprogramming[18], we decided to initiate reprogramming with a low seeding density, which would allow input cells to go through more cell cycles. Finally, our ultimate goal was to develop a reprogramming protocol that was clinically relevant; therefore, we focused on optimizing feeder-free plating conditions.

We initially pre-screened the mod-mRNA reprogramming protocols that utilized feeder-free plating conditions and eventually selected one which used a modified version of OCT4 fused with the MyoD transactivation domain (called M$_3$O)[19] in combination with five other reprogramming factors (SOX2, KLF4, cMYC, LIN28A, and NANOG)[15]. This 6-factor mod-mRNA reprogramming cocktail is referred to as 5fM$_3$O mod-mRNAs (Supplementary Fig. 1a). Transfecting this 5fM$_3$O mod-mRNA cocktail as previously described[15] resulted in a reprogramming efficiency of <0.5% (Supplementary Fig. 1b, c), which is consistent with published reports on mod-mRNA reprogramming[6,14,15]. When fibroblasts were plated at a low seeding density, we observed substantial cytotoxicity and cell death within 4–5 days of initiating mod-mRNA reprogramming (Supplementary Fig. 1b, c). Similar cytotoxicity and cell death was observed when the 5fM$_3$O mod-mRNA cocktail was supplemented with m-miRNAs (see Methods section).

To identify conditions that would support the growth of low-density fibroblast cultures during co-transfections with 5fM$_3$O mod-mRNAs and m-miRNAs, we screened various reprogramming media and RNA transfection reagents. We determined that RNA transfected with Lipofectamine RNAiMAX (RNAiMAX) at 24 h intervals allowed for the survival of low-density fibroblasts (500 cells per well of a 6-well format dish) cultured in the knock-out serum replacement (KOSR) reprogramming medium. However, despite their survival, the cells did not undergo complete reprogramming. We hypothesized that reprogramming failed due to inefficient mod-mRNA transfection (Fig. 1a and Supplementary Fig. 2; see Supplementary Figs. 2 and 3a for the results obtained on an independent human primary neonatal fibroblast line, FN1). To optimize the transfection efficiency of cells grown in KOSR medium, we used mod-mRNA encoding mWasabi to evaluate several different transfection buffers in combination with assessing the effect of adjusting the pH of the buffers. The highest transfection efficiency using the manufacturer's recommended transfection buffer for RNAiMAX, Opti-MEM, was achieved when its pH was adjusted from the original 7.2–7.3 to 8.2 (up to ~65% of mWasabi positive cells) (Fig. 1a and Supplementary Figs. 2 and 3a). We also found that phosphate-buffered saline (PBS) could be used as an alternative transfection buffer for RNAiMAX (Fig. 1a and Supplementary Fig. 3a). Interestingly, altering the pH or composition of transfection buffers did not affect the transfection efficiency of fluorescently labeled AllStars Negative small interfering RNA (siRNA), which was used as a control for the transfection of miRNA mimics, probably due to intrinsic properties of these RNAs (Supplementary Fig. 4). Therefore, to streamline the reprogramming procedure, m-miRNAs were delivered using the same transfection buffers as mod-mRNAs in the rest of our studies.

Employing Opti-MEM adjusted to a pH of 8.2 (Opti-MEM-8.2) as the transfection buffer for RNAiMAX, we reassessed the reprogramming of neonatal fibroblasts cultured in KOSR medium using mod-mRNAs and m-miRNAs transfections. The most efficient, consistent, and cost-effective mod-mRNA/m-miRNA reprogramming regimen is depicted in Fig. 1b. It involves seven transfections of 600 ng of 5fM$_3$O mod-mRNA cocktail and 20 pmol of m-miRNAs per well of a 6-well format dish performed every 48 h using Opti-MEM-8.2 as the transfection buffer. This protocol enabled us to achieve an ultra-high reprogramming efficiency yielding up to 4,019 (3,896 ± 131.14; mean ± s.d.; $n = 3$) (Fig. 1c, d and Table 1) and 3,391 (3,132 ± 240.04; mean ± s.d.; $n = 3$) (Supplementary Fig. 3b, c and Table 1) TRA-1-60-positive colonies from 500 initially plated cells using

two independent primary neonatal fibroblast lines, FN2 and FN1, respectively. These reprogramming efficiencies were higher than the efficiencies obtained with the previously published mod-mRNA reprogramming protocol[15] using the same fibroblast lines (Supplementary Fig. 1b, c; 50,000 cells plated per well) ($P < 0.0001$ for both FN1 and FN2 using the unpaired two-tailed Student's $t$test).

We found that three transfections performed every 48 h were the minimum required to obtain iPSC colonies, and transfections performed every 72 h showed a reduced capacity to generate iPSCs as compared to 48 h transfection intervals (Fig. 2). No

TRA-1-60-positive colonies arose when regular, unadjusted Opti-MEM at pH 7.3 (Opti-MEM-7.3) was used as the transfection buffer (Fig. 1c, d and Supplementary Fig. 3b, c). Reprogramming also failed when Opti-MEM at pH 8.6 was used as the transfection buffer despite the relatively high efficiency of mWasabi mRNA transfection achieved under this pH (Fig. 1a, c, d and Supplementary Fig. 3a–c). The regimen performed with Opti-MEM at pH 8.6 appeared to be cytotoxic, probably due to the degradation of RNA at this higher pH. Degraded RNA most likely increases the innate immunity response, which in turn induces cytotoxicity and reduces the reprogramming efficiency.

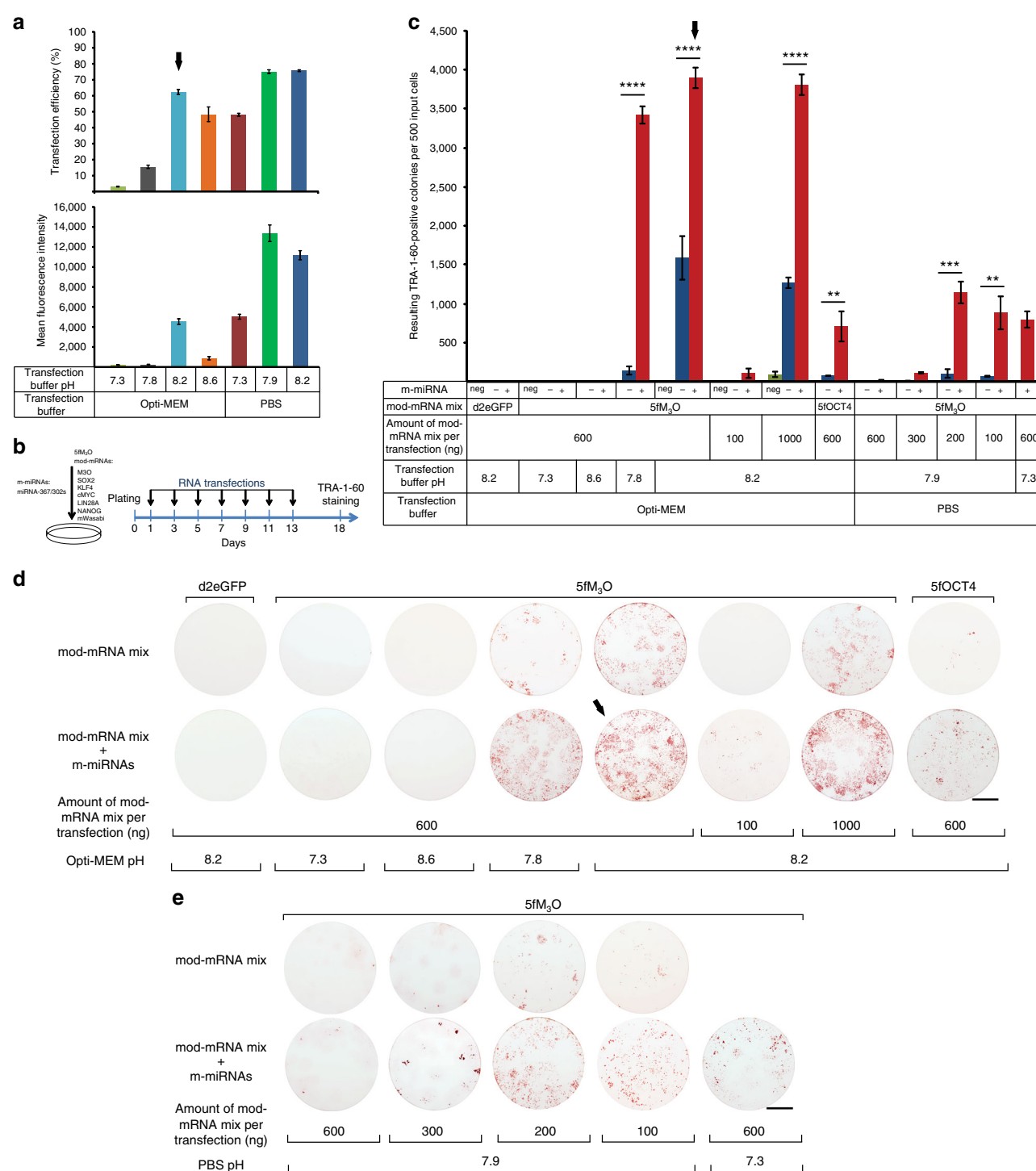

Varying the concentrations of m-miRNAs in combination with 600 ng of 5fM$_3$O mod-mRNAs did not lead to a substantial change in reprogramming efficiency (Fig. 3). Transfections every 48 h with m-miRNAs in combination with d2eGFP as a negative control failed to induce iPSC formation (Fig. 1c, d and Supplementary Fig. 3b, c). Interestingly, while the transfection of 5fM$_3$O mod-mRNAs alone at 48 h intervals resulted in a high yield of TRA-1-60-positive colonies (Fig. 1c, d and Supplementary Fig. 3b, c), we were not successful in establishing long-lived iPSC lines from these colonies. This suggests that our transfection regimen requires both m-miRNAs and 5fM$_3$O mod-mRNAs to produce stably reprogrammed iPSCs. Note that substituting M$_3$O with wild-type OCT4 in the 6-factor cocktail (5fOCT4) resulted in a marked reduction of the reprogramming efficiency (Fig. 1c, d and Supplementary Fig. 3b, c). Therefore, only the 5fM$_3$O mod-mRNA cocktail was used in the rest of our studies.

When 5fM$_3$O mod-mRNAs were delivered alone or in combination with m-miRNAs every 24 h, the high cytotoxicity of this regimen prevented the reproducible generation of iPSCs. This cytotoxicity likely resulted from an excess of exogenous mod-mRNAs, which is known to cause the activation of a cellular immune response[14]. Consistent with the high cytotoxicity of mod-mRNAs, a higher transfection efficiency of mod-mRNAs achieved with PBS at pH 7.9 as compared to Opti-MEM-8.2 (Fig. 1a and Supplementary Fig. 3a) did not translate into more efficient reprogramming due to poor survival of input cells (Fig. 1c, e and Supplementary Fig. 3b, d). Instead, the amount of 5fM$_3$O mod-mRNAs delivered with PBS at pH 7.9 had to be reduced to improve the survival and reprogramming efficiency of fibroblasts (Fig. 1c, e and Supplementary Fig. 3b, d).

Collectively, our results indicate that the fine-tuning of exogenous reprogramming mod-mRNAs levels in combination with m-miRNAs synergistically improves the efficiency of iPSC generation from human primary fibroblasts.

**Low density of input cells improves reprogramming efficiency.** To address how the initial seeding density of human primary cells affects the reprogramming efficiency of our optimized RNA-based approach (Fig. 1b), we performed a series of reprogramming experiments using human primary neonatal fibroblasts plated at different densities (Fig. 4a; see Supplementary Fig. 5 for the results obtained on an independent human primary neonatal fibroblast line, FN1). We found that the highest initial seeding density that allowed for the efficient generation of TRA-1-60-positive colonies was 10,000 cells per well in a 6-well format dish. The efficiency of reprogramming gradually increased with reducing the starting density of input cells (Fig. 4a and Supplementary

Fig. 5). Colonies formed poorly at an initial seeding density of 50,000 cells per well and above due to cell overcrowding in the process of reprogramming.

Reprogramming efficiency is traditionally calculated as the number of iPSC colonies generated divided by the number of input cells. If this method is used to calculate the efficiency of our optimal reprogramming regimen, an efficiency of ~800% is obtained. This high efficiency suggests that multiple sister iPSC colonies must be derived from a single parental cell. To address the formation of multiple sister iPSC colonies, we performed a single-cell reprogramming experiment. We were able to reprogram up to 90.7% of individually plated single cells (Fig. 4b), with the majority of input cells producing multiple TRA-1-60-positive colonies (Fig. 4b, c). If m-miRNAs were excluded from the regimen, the efficiency of single-cell reprogramming dropped drastically (Fig. 4b), further demonstrating the synergism between reprogramming mod-mRNAs and m-miRNAs on the efficiency of iPSC generation.

**The RNA-based approach reprograms a variety of fibroblasts.** Since our goal was to develop a clinically relevant protocol, we evaluated the applicability of our approach for the reprogramming of human primary fibroblasts derived from a variety of human subjects. In addition to three neonatal fibroblast lines, we successfully reprogrammed fibroblasts derived from patients with inherited skin blistering disorders and Down syndrome, as well as from three healthy adult individuals of 40 (F40), 50 (F50), and 62 (F62) years of age with high, albeit different, efficiencies (Table 1 and Supplementary Fig. 6). The protocol is more cytotoxic to adult and especially disease-specific lines, probably due to the activation of senescence-associated pathways in these cells[20]. Therefore, the reprogramming of adult cells was initiated at higher cell numbers and required adjusting plating densities based on the patient's age (Table 1).

As a more stringent test of the robustness of our reprogramming protocol, we assessed the ability of the method to reprogram senescent fibroblasts. The F50 fibroblast line was serially passaged until more than 91% of cells exhibited a senescent phenotype (Fig. 5a). The reprogramming of this senescent line took only 16 days and resulted in an efficiency of ~0.33% (Table 1 and Supplementary Fig. 6), which surpassed the reprogramming efficiency previously reported for senescent fibroblasts using an integrating lentiviral approach[21]. To our knowledge, this is the first report of the successful reprogramming of senescent human cells with an integration-free approach. The iPSCs derived from these senescent fibroblasts exhibited the expected markers of rejuvenation, including the downregulation of p21

**Fig. 1** Optimal delivery of mod-mRNAs and m-miRNAs enhances the reprogramming of human primary fibroblasts. **a** Transfection efficiency (top) and mean fluorescence intensity (bottom) of human primary neonatal fibroblasts (FN2) transfected with 500 ng of mod-mRNA encoding mWasabi, using indicated transfection buffers, as determined by flow cytometry 24 h post transfection. Error bars, mean ± s.d. for all panels ($n = 3$). The results are reproducible using an independent primary neonatal fibroblast line, FN1 (Supplementary Fig. 3a). **b** Schematic diagram of the optimized RNA-based reprogramming regimen with RNAs delivered using Opti-MEM adjusted to a pH of 8.2 as the transfection buffer for RNAiMAX. **c** Effect of mod-mRNA titration and the addition of m-miRNAs on the reprogramming of human primary neonatal fibroblasts (FN2). All reprogramming conditions were initiated at 500 cells per well of a 6-well format dish. Cells were transfected every 48 h with differing amounts of mod-mRNAs encoding mWasabi (transfection control) and either d2eGFP as a negative control or 6-factor reprogramming cocktails containing either M$_3$O (5fM$_3$O) or OCT4 (5fOCT4). Mod-mRNA transfections were performed alone or in combination with m-miRNAs or AllStars Negative control siRNA (neg) transfections, using the indicated transfection buffers. The number of resulting TRA-1-60-positive colonies on day 18 of the indicated regimens are plotted. Error bars, mean ± s.d. ($n = 3$). The yield of TRA-1-60-positive colonies was compared between the regimens performed in the presence or absence of m-miRNAs. $P$ values were calculated using the unpaired two-tailed Student's $t$ test. **$P < 0.01$, ***$P < 0.001$, ****$P < 0.0001$. **d** Representative TRA-1-60-stained reprogramming wells corresponding to conditions indicated in **c** for Opti-MEM as the transfection buffer. **e** Representative TRA-1-60-stained reprogramming wells corresponding to conditions indicated in **c** for PBS as the transfection buffer. Solid black arrows indicate optimal conditions for mod-mRNA transfections in **a** and iPSC colony generation in **c**, **d**. All scale bars, 10 mm. The results of reprogramming experiments are reproducible using an independent primary neonatal fibroblast line, FN1 (Supplementary Fig. 3b–d)

**Table 1 Generation of iPSCs from a variety of fibroblast lines using the RNA-based approach**

| Fibroblast line | Age (years old) | Input cells per well | TRA-1-60-positive colonies per well | Efficiency (%) |
|---|---|---|---|---|
| FN1 | Neonatal | 500 | 3132 ± 240.04 | 626.4 ± 48.01 |
| FN2 | Neonatal | 500 | 3896 ± 131.14 | 779.2 ± 26.23 |
| FN5 | Neonatal | 500 | 2161.7 ± 258.8 | 432.3 ± 51.76 |
| F40 | 40 | 2000 | 1453.3 ± 93.33 | 72.67 ± 4.67 |
| F62 | 62 | 3000 | 1828 ± 201.5 | 60.93 ± 6.72 |
| F50 | 50 | 5000 | 1821.7 ± 90.5 | 36.43 ± 1.81 |
| F50S | 50 (senescent) | 100,000 | 325 ± 88.66 | 0.33 ± 0.09 |
| FD54 | Neonatal | 3000 | 129.3 ± 52.62 | 4.31 ± 1.76 |
| FEH1 | 7 | 1000 | 405.7 ± 14.57 | 40.57 ± 1.46 |
| FEB1 | 23 | 3000 | 363.7 ± 44.5 | 12.1 ± 1.5 |
| FRD1 | 21 | 3000 | 125.7 ± 33.13 | 4.2 ± 1.1 |

Primary neonatal fibroblast lines: FN1, FN2, and FN5
Healthy primary adult fibroblast lines: F40, F62, and F50
A senescent fibroblast line derived from F50: F50S
Disease-associated fibroblast lines: Down syndrome (FD54), epidermolytic ichthyosis (FEH1), epidermolysis bullosa simplex (FEB1), and recessive dystrophic epidermolysis bullosa (FRD1)
Values are reported as mean ± s.d. ($n = 3$)
iPSC Induced pluripotent stem cell

(Supplementary Fig. 7), reactivation of telomerase (Supplementary Fig. 8), and elongation of telomeres (Fig. 5b).

**Characterization of the generated iPSC lines**. Multiple iPSC lines were derived from neonatal, adult, and senescent human fibroblasts using the optimized RNA-based approach (Supplementary Table 1). All established lines exhibited appropriate karyotypes (Supplementary Table 1 and Supplementary Fig. 9) and showed molecular and functional characteristics of pluripotent stem cells. The transcriptional profiles of selected iPSC lines (Supplementary Table 2) appeared to be similar to those of human ESC lines (Fig. 6a and Supplementary Fig. 10). The generated iPSC lines clustered closely to the human ESC lines H1 and H9 when the global transcriptional profiles were subjected to the principal components analysis (Fig. 6a). The expression of the pluripotency markers OCT4, NANOG, SOX2, LIN28A, TRA-1-81, and SSEA-4 was validated at the protein level (Figs. 5c and 6b and Supplementary Figs. 11, 12), and the demethylation of the OCT4 promoter was confirmed by bisulfite sequencing (Fig. 6c).

The generated iPSCs successfully underwent directed differentiation into βIII-Tubulin (TUJ1)-positive neurons (ectoderm) and cytokeratin Endo-A-positive endodermal cells (Figs. 5d and 6d), and produced mesoderm-derived beating cardiomyocytes from embryoid bodies (Supplementary Movies 1–5). The developmental potential of the generated iPSCs was further confirmed in vivo by the formation of teratomas that consisted of cell types of all three germ layers (Figs. 5e and 6e and Supplementary Fig. 13).

**Gene expression changes in cells undergoing reprogramming**. To address the potential mechanisms behind the increased reprogramming efficiency observed in our protocol vs. the conventional 5fM$_3$O mod-mRNA reprogramming protocol[15], we analyzed the transcript levels of a selected set of genes at different time points during the first 16 days of reprogramming using the Nanostring nCounter Gene Expression Assay. Two independent human primary neonatal fibroblast lines, FN1 and FN2, were subjected to a time-course gene expression analysis during reprogramming performed with different regimens as indicated in Supplementary Fig. 14. Our optimized RNA-based approach (called 5fM$_3$O + m-miRNAs) initiated at 500 cells per well showed lower levels of transcripts encoding innate immunity genes than the previously published feeder-free 5fM$_3$O mod-mRNA reprogramming protocol[15] (called control 5fM$_3$O

reprogramming) (FN2: Fig. 7a and Supplementary Fig. 15a; FN1: Supplementary Figs. 16a and 17a). This finding correlated with a lower level of exogenous reprogramming mod-mRNAs detected in the 5fM$_3$O + m-miRNA regimen (FN2: Fig. 7a and Supplementary Fig. 15a; FN1: Supplementary Figs. 16a and 17a) and presumably contributed to the improved reprogramming efficiency. The 5fM$_3$O + m-miRNA reprogramming initiated at 500 cells per well yielded higher expression levels of several cell cycle promoting genes as compared to the 5fM$_3$O + m-miRNA reprogramming initiated at higher seeding densities (10,000 or 50,000 cells per well) or the control 5fM$_3$O reprogramming (FN2: Fig. 7b and Supplementary Fig. 15b; FN1: Supplementary Figs. 16b and 17b). The activation of cell cycle genes correlated with the cell expansion rate for all assessed regimens for both cell lines (FN2: Fig. 7b; FN1: Supplementary Fig. 16b). The expression of both p21$^{CIP1}$ and p57 remained low in the 5fM$_3$O + m-miRNA approach as compared to the control 5fM$_3$O reprogramming (FN2: Fig. 7b; FN1: Supplementary Fig. 16b) and may be attributed to optimal levels of expression of reprogramming factors.

We also analyzed transcript levels of genes known to be involved in chromatin remodeling and pluripotency maintenance with a particular focus on known predictive markers of pluripotency such as UTF1, LIN28A, DPPA2, and SOX2[22] (FN2: Fig. 7c and Supplementary Fig. 15c; FN1: Supplementary Figs 16c and 17c). The activation of the majority of chromatin remodeling and pluripotency genes occurred several days earlier in the 5fM$_3$O + m-miRNA approach as compared to the control 5fM$_3$O reprogramming (FN2: Fig. 7c and Supplementary Fig. 15c; FN1: Supplementary Figs 16c and 17c). Thus, the optimized RNA-based approach not only reduces the expression of innate immunity genes, but also leads to the robust activation of pluripotency-associated genes.

**Discussion**
In this study, we developed a highly optimized, non-integrating, combinatorial RNA-based reprogramming approach that allowed us to reprogram multiple normal and disease-specific human fibroblast lines into iPSCs at an ultra-high efficiency. The approach is cost effective, provides an opportunity to shorten the time between the biopsy and the generation of clinically relevant, high-quality iPSC lines, and allows for the production of iPSCs from as few as a single cell in a feeder-free system (Fig. 4b, c). The high reprogramming efficiency of this approach, in combination

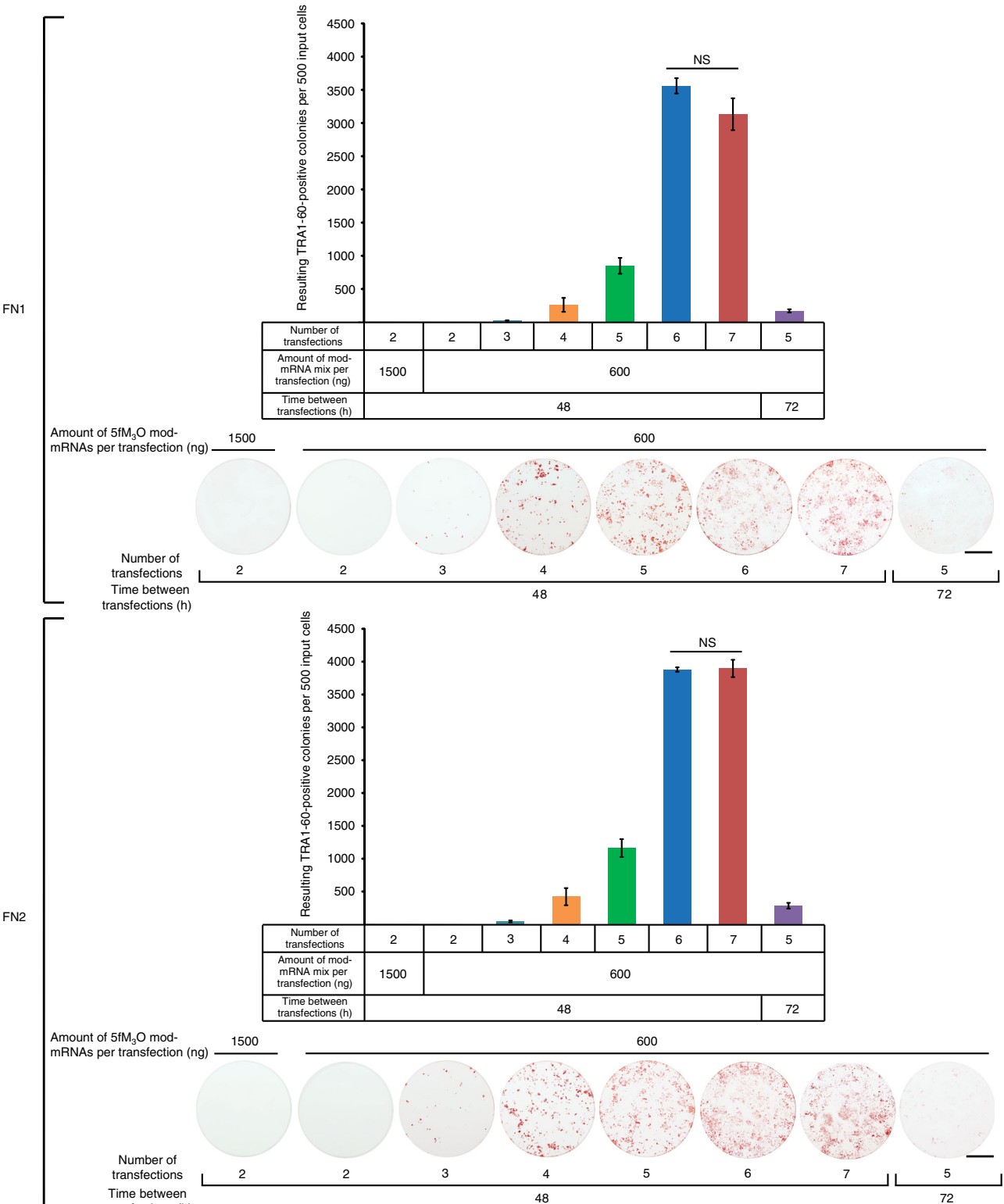

**Fig. 2** Defining the minimal and the optimal number of RNA transfections for the RNA-based reprogramming approach. Summary plots and representative TRA-1-60-stained reprogramming wells show the yield of TRA-1-60-positive colonies at day 18 of reprogramming initiated at a plating density of 500 cells per well of a 6-well format dish with two independent human primary neonatal fibroblast lines, FN1 (top) and FN2 (bottom). The reprogramming was accomplished using the indicated numbers of mod-mRNA and m-miRNA transfections performed at either 48 or 72 h intervals using Opti-MEM-8.2 as the transfection buffer. The amount of m-miRNA used per transfection is 20 pmol, and the amount of mod-mRNAs is indicated. Error bars, mean ± s.d. ($n = 3$). No statistical difference in the reprogramming efficiencies was observed between 6 and 7 transfections (NS, $P > 0.05$). $P$ values were calculated using the unpaired two-tailed Student's $t$ test. Scale bars, 10 mm

| Fibroblast lines (500 input cells) | m-miRNAs per transfection | TRA-1-60-positive colonies/well | Efficiency (%) |
|---|---|---|---|
| FN1 | 10 pmoles | 3608.3 ± 67.57 | 721.67 ± 13.51 |
| | 40 pmoles | 3036.3 ± 101.01 | 607.27 ± 20.2 |
| FN2 | 10 pmoles | 3915 ± 68.46 | 783 ± 13.69 |
| | 40 pmoles | 3242.7 ± 66.56 | 648.53 ± 13.31 |

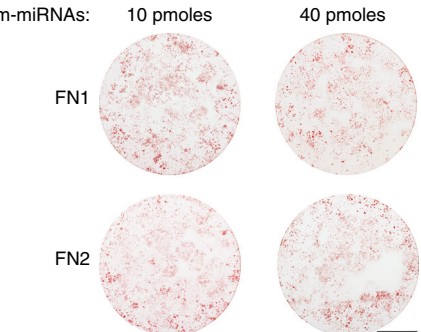

**Fig. 3** The efficiency of the RNA-based reprogramming approach using different m-miRNA concentrations. Summary table and representative TRA-1-60-stained reprogramming wells show the yield of TRA-1-60-positive colonies at day 18 of the optimized RNA-based reprogramming regimen (Fig. 1b) performed with differing amounts of m-miRNAs as indicated. The protocol was initiated with two independent human primary neonatal fibroblast lines (FN1 and FN2) plated at a density of 500 cells per well of a 6-well format dish. The reprogramming efficiency was calculated by dividing the number of resulting TRA-1-60-positive colonies by the number of input cells and multiplying by 100%. Mean ± s.d. ($n = 3$). Scale bar, 10 mm

with the small number of input cells, may especially benefit the clinical application of iPSCs as it potentially reduces the accumulation of mutations that arise in iPSCs due to extensive culturing of patient's cells or selective reprogramming of mutated founder cells. Our results also suggest that the reprogramming efficiency for other cell types may be greatly improved by optimizing protocols in a cell-type-specific manner.

The high reprogramming efficiency observed in our optimized protocol may seem at odds with previous reports that show only a modest improvement in reprogramming efficiency following the combined treatment with mod-mRNAs and pluripotency-inducing miRNAs[6,7]. Here, we enhanced the efficiency of a combinatorial mod-mRNA/miRNA-based reprogramming strategy by optimizing the transfection efficiency of reprogramming factors, carefully titrating the amount of mod-mRNAs, and identifying the optimal cell culture conditions that would allow for the maximum proliferation of fibroblasts (see the model in Fig. 8).

First, we were able to initiate reprogramming at a low density under feeder-free conditions, addressing a known limitation of mod-mRNA-based reprogramming that requires a substantial number of input cells for successful iPSC generation[6,15]. The initial low seeding density encouraged input cells to go through more cell cycles during reprogramming, potentially allowing for more efficient chromatin remodeling and improved reprogramming efficiency (Fig. 7b and Supplementary Fig. 16b). In agreement with this assumption, the expression of ASF1A, a histone-remodeling protein important for cellular reprogramming, remained high throughout our protocol and correlated with the cell expansion rate (Fig. 7c and Supplementary Fig. 16c).

Second, we reduced the cytotoxic and immunogenic effect of repetitive mod-mRNA transfections by fine-tuning the

transfection regimen and by supplementing the reprogramming process with miRNA-367/302s as mature miRNA mimics. The addition of m-miRNAs appears to be critical for the reduction of exogenous mod-mRNA levels to the optimum capable of inducing efficient reprogramming. The exclusion of m-miRNAs from the protocol reduced the efficiency of reprogramming and affected the stability of the resulting iPSCs. Our approach requires less mod-mRNAs per transfection and performs more consistently with every other day transfections. As a result, the regimen prevents the induction of a robust innate immunity response as shown by the low activation of innate immunity genes (Fig. 7a and Supplementary Figs. 15a, 16a, and 17a). Nevertheless, the commonly used inhibitor of interferon response B18R was still used in our optimized protocol (see Methods section). While the current trend in the field is to increase the delivery or expression levels of reprogramming factors to improve the efficiency of iPSC generation, we have shown that efficient reprogramming can be achieved by balancing rather than simply enhancing the expression level of factors. An example of balancing the efficiency of transfection with the amount of mod-mRNAs is shown in the experiment where we used PBS (pH 7.9) as the transfection buffer. In this experiment, the reprogramming efficiency was improved by either reducing the amount of 5fM$_3$O mod-mRNAs used per transfection from 600 to 200 ng or by decreasing the pH of PBS to 7.3 (Fig. 1c, e and Supplementary Fig. 3b, d), which consequently decreased the transfection efficiency of mod-mRNAs (Fig. 1a and Supplementary Fig. 3a).

The combination of a reprogramming mod-mRNA cocktail with m-miRNAs not only prevented the robust activation of an innate immunity response, but also appeared to have a synergistic effect on reprogramming. Although the synergism between reprogramming mod-mRNAs and m-miRNAs has been previously suggested[6,7], it has never been shown at such a dramatic level as reported here. This synergistic effect can be especially appreciated in our single-cell reprogramming experiments, where the exclusion of m-miRNAs from the reprogramming cocktail drastically diminished the ability of individually plated cells to undergo reprogramming (Fig. 4b). This synergism is most likely mediated by the ability of reprogramming m-miRNAs to increase cell cycling and to target multiple pluripotency-associated and differentiation-associated pathways that overlap with the action of reprogramming transcription factors. As an example, we showed that the chromatin modifier TET1 and the strong inducer of pluripotency SALL4 were both robustly activated in response to the delivery of m-miRNAs alone. These genes exhibited even faster kinetics of activation in our optimized protocol (Fig. 7c and Supplementary Fig. 16c). SALL4 has previously been shown to promote an ordered activation of ESC-specific miRNAs, which is associated with a higher reprogramming efficiency[23]. Therefore, our findings suggest a role for SALL4 in mediating the synergism between mod-mRNAs and m-miRNAs during reprogramming.

Our optimized regimen reprograms up to 90.7% of individually plated human primary neonatal fibroblasts (Fig. 4b) and generates iPSCs from primary neonatal and adult human fibroblasts with a 100% success rate, albeit with variable efficiencies (Table 1). Despite these results, the reprogramming process is not completely penetrant. Many non-reprogrammed TRA-1-60-negative cells remain at the end of our protocol despite the high number of iPSC colonies generated. Since rapid cell division may cause continuous dilution of exogenous reprogramming factors and prevent the maintenance of reprogramming factor expression at levels sufficient to achieve simultaneous reprogramming in all founder cells, deterministic reprogramming may be difficult to achieve with an RNA-based approach. Interestingly, while our method reprogramed patient's fibroblasts with a high efficiency, we were not successful in adapting our protocol to

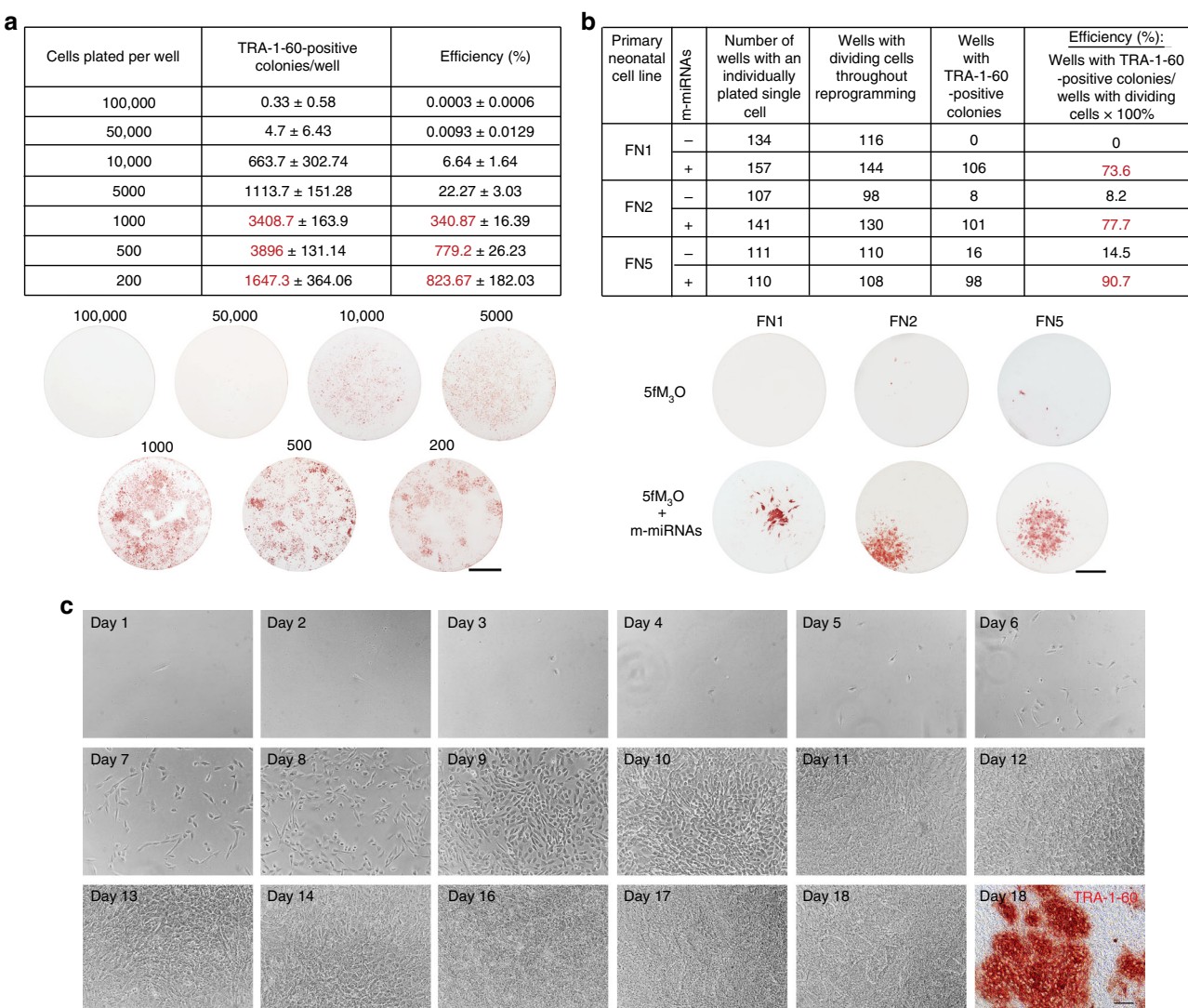

**Fig. 4** Low initial cell plating densities improve the efficiency of the RNA-based reprogramming approach. **a** Summary table and representative TRA-1-60-stained reprogramming wells showing the yield of TRA-1-60-positive colonies at day 18 of the optimized RNA-based reprogramming regimen (see Fig. 1b) initiated with human primary neonatal fibroblasts (FN2) at the indicated cell plating numbers per well of a 6-well format dish. The efficiency was calculated by dividing the number of resulting TRA-1-60-positive colonies by the number of input cells and multiplying by 100%. Mean ± s.d. (n = 3). Scale bar, 10 mm. Similar results were obtained using an independent primary neonatal fibroblast line, FN1 (Supplementary Fig. 5). **b** Summary table showing the reprogramming of individually plated single cells with the 5fM₃O mod-mRNA reprogramming cocktail delivered alone or in combination with m-miRNAs every 48 h using Opti-MEM adjusted to a pH of 8.2 as the transfection buffer (see Fig. 1b). The number of individually plated single cells from three independent primary neonatal fibroblast lines (FN1, FN2, and FN5), the survival of these cells throughout reprogramming, and the resulting number of wells with at least one TRA-1-60-positive colony at day 18 of reprogramming are shown. The efficiency was calculated by dividing the number of wells with TRA-1-60-positive colonies by the number of wells with surviving input cells and multiplying by 100%. Representative TRA-1-60-stained reprogramming wells (48-well dish format) correspond to conditions from the summary table. Note the formation of multiple sister TRA-1-60-positive colonies when both mod-mRNA and m-miRNA transfections were employed. Scale bar, 3 mm. **c** Representative day-by-day images of single-cell reprogramming (FN2) performed with the optimized RNA-based reprogramming approach. Staining for TRA-1-60 performed on day 18 of reprogramming is also shown. Scale bar, 100 μm

reprogramming the established BJ fibroblast line. This suggests that established, long-lived fibroblast lines, which are commonly used as reference lines for reprogramming protocols, may not be an ideal representative of human primary fibroblasts' behavior during reprogramming.

In conclusion, the ultra-high reprogramming efficiency of our integration-free approach addresses the current limitations of reprogramming techniques for potential clinical applications and for studying the mechanisms of human somatic cell reprogramming. Our results also emphasize the importance of finding the optimal cell-type-specific conditions to reveal the full synergistic potential of exogenous mod-mRNAs and miRNAs in reprogramming somatic cells.

## Methods

**Generation of IVT templates.** Production of in vitro transcription (IVT) templates for mod-mRNA generation was adapted from Warren et al[14]. Briefly, plasmids used as template for PCR for IVT template preparation were obtained from Addgene: pcDNA3.3_KLF4 (catalog # 26815); pcDNA3.3_OCT4 (catalog # 26816); pcDNA3.3_SOX2 (catalog # 26817); pcDNA3.3_c-MYC (catalog # 26818); pcDNA3.3_LIN28A (catalog # 26819); pcDNA3.3_d2eGFP (catalog # 26821). Additional plasmids with inserts encoding *mWasabi*, human *NANOG*, and *M₃O* (pcDNA3.3_mWasabi, pcDNA3.3_NANOG, pcDNA3.3_M3O) used for IVT

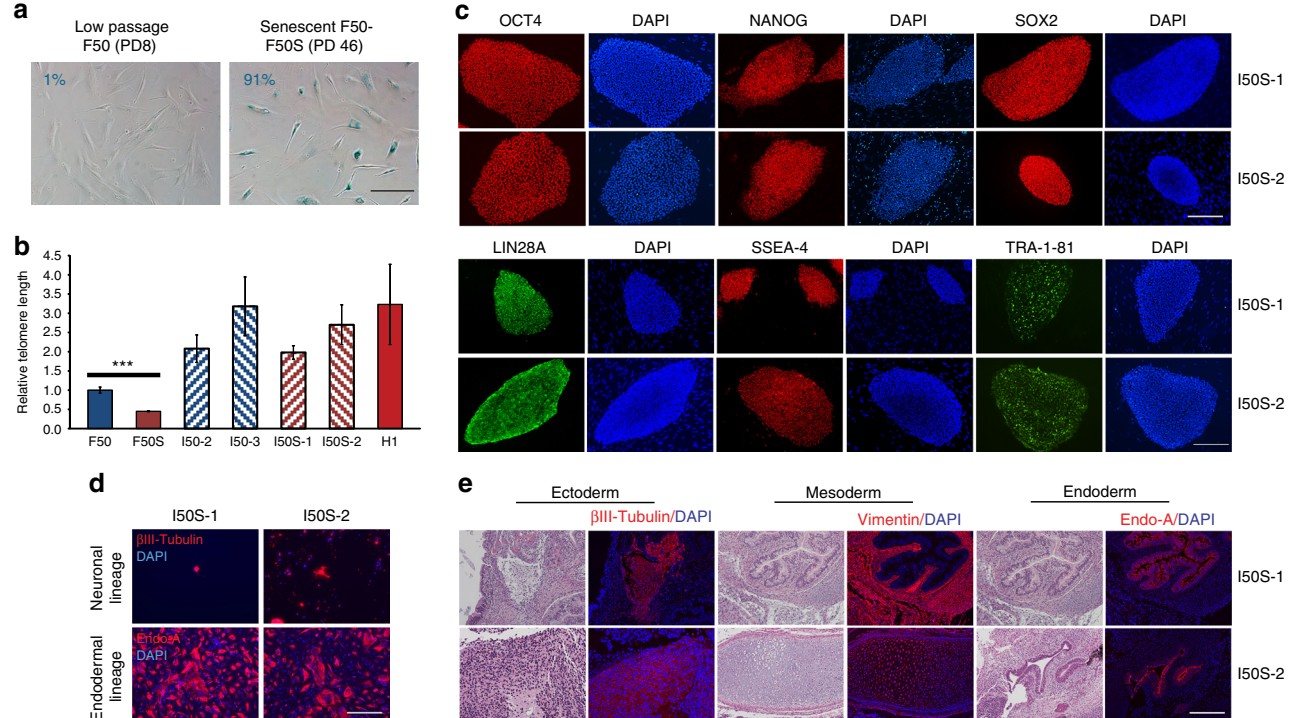

**Fig. 5** Successful reprogramming of senescent human fibroblasts. **a** Adult human primary fibroblasts from a 50-year-old patient (F50) were passaged until they reached 46 population doublings (PDs), at which point the cells stopped dividing, and started exhibiting a senescent phenotype with an enlarged cellular morphology and 91% positivity for senescence-associated β-galactosidase (F50S). Scale bar, 200 μm. **b** Elongation of telomeres in iPSCs generated from F50 and F50S. The length of telomeres in the indicated cell lines was measured by qPCR analysis. Error bars, mean ± s.d. ($n = 3$). Prolonged fibroblast passaging of F50 resulted in a significant shortening of telomeres in F50S (***$P < 0.001$). $P$ values were calculated using the unpaired two-tailed Student's $t$ test. **c** Immunofluorescent analysis showing expression of a panel of pluripotency markers in iPSC lines derived from F50S (I50S-1 and I50S-2). See Fig. 6b and Supplementary Figs. 11 and 12 for controls and iPSC lines derived from F50. Scale bar, 250 μm. **d** Immunostaining showing expression of the neuronal marker βIII-Tubulin (TUJ1) (ectoderm), and the endoderm-specific cytokeratin Endo-A in iPSCs derived from F50S subjected to directed differentiation. See Fig. 6d for directed differentiation of iPSC lines derived from F50. Scale bar, 250 μm. **e** Hematoxylin and eosin staining and immunofluorescent analysis of consecutive sections of teratomas derived from I50S-1 and I50S-2 showing histology and marker expression specific to ectoderm (βIII-Tubulin (TUJ1), neural tissues), mesoderm (vimentin, connective tissues), and endoderm (Endo-A, endothelium). See Fig. 6e for iPSC lines derived from F50. Scale bar, 250 μm

template preparation were cloned using the following Addgene plasmids: pTEC15 (mWasabi, catalog # 30174); pGEM-NANOG (catalog # 16351); pMXs-hM₃O-IP (catalog # 46645). IVT templates were PCR generated using listed plasmids as templates. Forward primer is 5′-TTGGACCCTCGTACAGAAGCTAATACG-3′ and reverse primer used to introduce 120 polyA tail sequence is 5′-TTTTTTT TTTTTTTTTTTTTTTTTTTTTTTTTTTTTTTTTTTTTTTTTTTTTTTT-TTTTTTTTTTTTTTTTTTTTTTTTTTTTTTTTTTTTTTTTTTTTTTTTT-TTTTTTTTTCTTCCTACTCAGGCTTTATTCAAAGACCA-3′. These primers were synthesized by Integrated DNA Technologies. Reverse primer was synthesized as Ultramer oligos at a 4 nmol scale.

**Preparation of mod-mRNAs and m-miRNAs**. Mod-mRNA was synthesized as described with slight modifications[14]. Specifically, 1.6 μg of template PCR product was provided for each 40 μl reaction of MEGAscript T7 Kit (Life Technologies). A 2.5× custom ribonucleoside mix including 15 mM 3′-0-Me-m⁷G(5′)ppp(5′)G ARCA cap analog (New England Biolabs), 3.75 mM guanosine triphosphate and 18.75 mM adenosine triphosphate (both were used from MEGAscript T7 Kit), and 18.75 mM 5-methylcytidine triphosphate and 18.75 mM pseudouridine triphosphate (TriLink Biotechnologies) was prepared. RNA synthesis reactions were incubated at 37 °C for 6 h and then treated with DNase for 15 min at 37 °C as directed by the manufacturer. RNA was purified with RNeasy Mini Kit columns (Qiagen) and then treated with Antarctic Phosphatase (New England Biolabs) for 30 min at 37 °C. After re-purification, RNA was eluted with nuclease-free dH₂O supplemented with 1 U/μl of RIBOGUARD™ RNase Inhibitor (Epicentre Biotechnologies). RNA was then quantitated by Nanodrop (Thermo Scientific) and stored at −80 °C until further use.

Unless otherwise noted, the mod-mRNA mix used for reprogramming ("reprogramming cocktail") contained 6 human reprogramming factors, M₃O (MyoD-Oct4)[19], SOX2, KLF4, cMYC, NANOG, and LIN28A (abbreviated as "5fM₃O mod-mRNAs"), at a molar stoichiometry of M₃O to the other 5 factors as 3:1:1:1:1:1 and included 10% mWasabi mod-mRNA to control for transfection

efficiency. A similar ratio of mod-mRNAs was maintained when OCT4 was used instead of M₃O in the 6-factor reprogramming cocktail (abbreviated as "5fOCT4 mod-mRNAs"). As a negative control for reprogramming (d2eGFP), the 6-factor reprogramming cocktail was substituted with the mod-mRNA mix containing 90% destabilized eGFP (d2eGFP) mod-mRNA and 10% mWasabi mod-mRNA. For reprogramming and transfection experiments, the mod-mRNA mix or mWasabi mod-mRNA alone were prepared at 100 ng/μl in nuclease-free water.

miRNA-367/302s as miScript miRNA mimics (Syn-has-miR-367-3p, Syn-has-miR-302a-3p, Syn-has-miR-302b-3p, Syn-has-miR-302c-3p, and Syn-has-miR-302d-3p) or miRNA negative controls (AllStars Negative Control siRNA and fluorescently labeled AllStars Negative siRNA AF 488) were purchased from Qiagen. Lyophilized products were dissolved at a 5 μM final concentration in nuclease-free water. Individual m-miRNA-367/302s stocks were mixed in 1:1:1:1:1 ratio to prepare a 5 μM combined m-miRNA reprogramming cocktail.

**Cells**. Three independent human primary neonatal fibroblasts lines (FN1, FN2, and FN5), two healthy adult primary fibroblasts lines (F50 and F40), and a neonatal fibroblast line from an individual with Down syndrome (FD54) were obtained from ATCC. The primary fibroblast cell line from a 62-year-old patient (F62) was obtained from Lonza. Fibroblasts from patients with inherited skin blistering diseases: epidermolytic ichthyosis with a heterozygous dominant p.[Asn188Ser] mutation in the *KRT1* gene (FEH1), severe generalized epidermolysis bullosa simplex with a heterozygous dominant p.[Arg125Cys] mutation in the *KRT14* gene (FEB1), and severe generalized recessive dystrophic epidermolysis bullosa (EB24) with a homozygous c.[6508C>T];[6508C>T], p.[Gln2170*];[p.Gln2170*] mutation in the *COL7A1* gene (FRD1)[24] were isolated from skin biopsies obtained with informed consent and with an approval from the local institutional review board committees (COMIRB and Medisch Ethische Commissie UMCG). H1 (WA01) and H9 (WA09) hESC lines were obtained from WiCell. A senescent fibroblast line (F50S) was obtained by serial passaging of F50 fibroblasts in fibroblast medium containing minimum essential medium (MEM) supplemented with 10% heat-

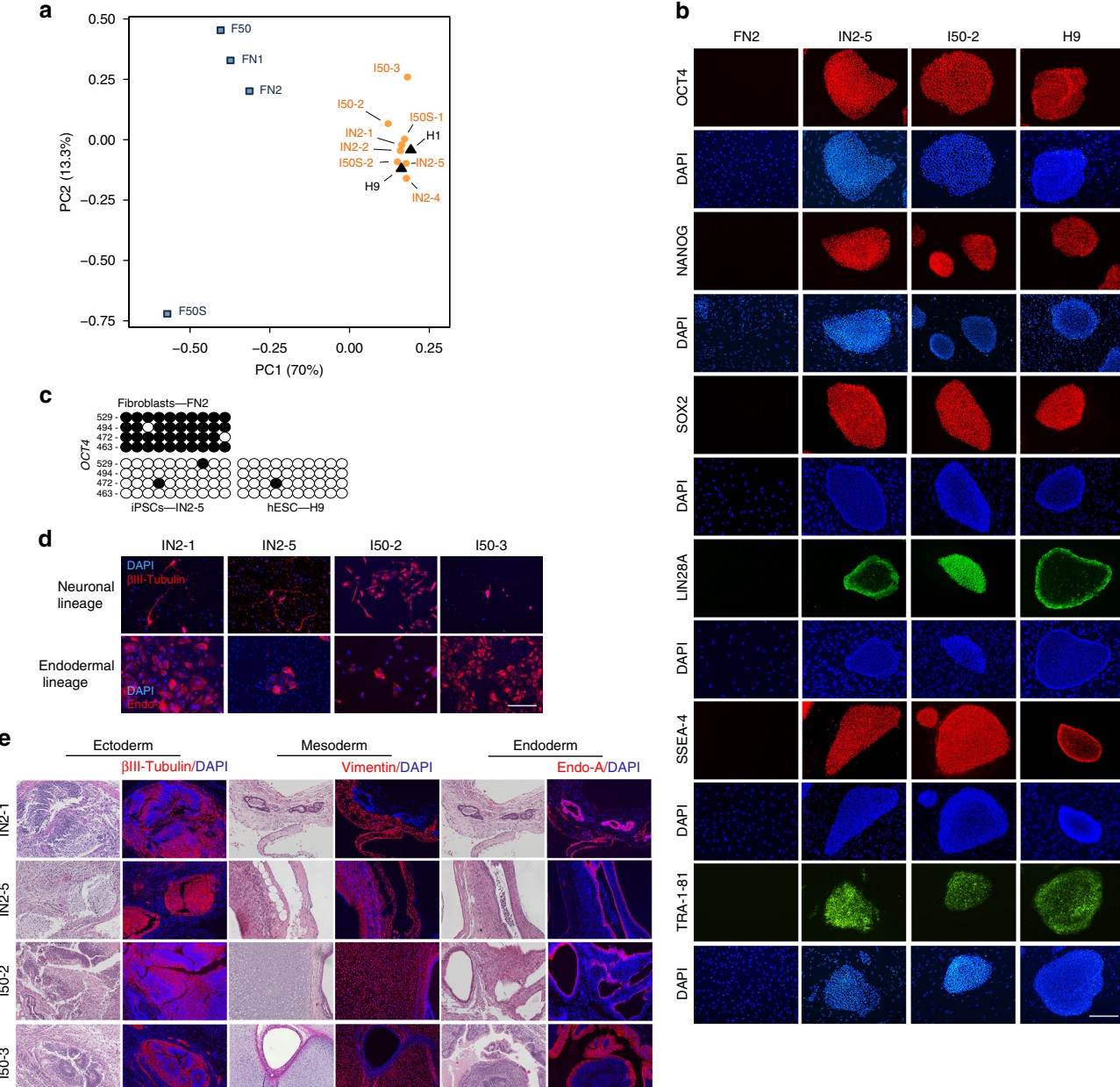

**Fig. 6** Characterization of the pluripotency of selected iPSC lines. **a** Principal component analysis of the RNA-Seq gene expression data performed on selected iPSC lines (orange circles) and their corresponding parental fibroblasts (blue squares), as well as on two human ESC lines (H1 and H9, black triangles), documenting the close clustering of iPSCs with H1 and H9. All IN2 iPSC lines were derived from FN2, I50 from F50, and I50S from F50S (see also Supplementary Table 1). **b** Immunofluorescent analysis showing expression of selected pluripotency markers in the indicated iPSC lines. H9 human ESCs and FN2 fibroblasts are included as positive and negative controls, respectively. See Fig. 5c and Supplementary Figs. 11 and 12 for the analysis of additional iPSC lines. **c** Bisulfite sequencing of the *OCT4* promoter region in a selected iPSC line and its parental fibroblast line. H9 human ESCs are included as a control. The closed circles indicate methylated sites of the analyzed region. **d** Immunostaining showing expression of the neuronal marker βIII-Tubulin (TUJ1) (ectoderm), and the endoderm-specific cytokeratin Endo-A in the indicated iPSC lines subjected to directed differentiation. **e** Hematoxylin and eosin staining and immunofluorescent analysis of consecutive sections of teratomas derived from indicated iPSC lines showing histology and marker expression specific to ectoderm (βIII-Tubulin (TUJ1), neural tissues), mesoderm (vimentin, connective tissues), and endoderm (Endo-A, endothelium). See Fig. 5e and Supplementary Fig. 13 for teratoma analysis of additional iPSC lines. All scale bars, 250 µm

inactivated fetal bovine serum (HI-FBS), 1× MEM non-essential amino acids solution, 55 µM of 2-mercaptoethanol (β-ME), 1× GlutaMAX™ supplement, plus antibiotics (all from Thermo Fisher Scientific), until the cells reached 46 population doublings (PDs). Senescence was confirmed by β-galactosidase activity using a Senescence β-Galactosidase Staining Kit (Cell Signaling Technology) according to the manufacturer's instruction.

Pluripotent stem cells were cultured in a 5% O₂/5% CO₂ tissue culture incubator and maintained either in N2B27 medium comprising a mixture of Dulbecco's modified Eagle's medium/F12 (DMEM/F12) and Neurobasal medium (Thermo Fisher Scientific) at a 1:1 ratio, 1× MEM non-essential amino acids solution, 1× GlutaMAX™ supplement, 55 µM β-ME, 1× N2 supplement (Thermo Fisher Scientific), 1× B27 supplement (Thermo Fisher Scientific), 50 µg/ml L-ascorbic acid (Sigma-Aldrich), 0.05% bovine serum albumin (BSA), 50 U/ml penicillin–streptomycin, and 100 ng/ml basic FGF (bFGF) (Thermo Fisher Scientific) on mitomycin C-inactivated human primary neonatal fibroblasts as a feeder or in E8 medium (Thermo Fisher Scientific) on tissue culture plates coated

with Geltrex® Matrix (Thermo Fisher Scientific) at 100× dilution according to the manufacturer's instruction. Y-27632 (Sigma-Aldrich) was used with each passaging at a final concentration of 10 μM to improve the survival of pluripotent stem cells. The medium was replaced daily. Y-27632 was removed the next day after each new passage, and iPSCs were cultured without Y-27632 until the next passage. All cell lines used in the study were tested negative for mycoplasma using Universal Mycoplasma Detection Kit (ATCC).

The cumulative PD level in Fig. 7b and Supplementary Fig. 16b is calculated by using the formula: PD = log($n_t$/$n_i$)/log 2, where $n_i$ is the initial number of cells and $n_f$ is the final number of cells at each time point described in Supplementary Fig. 14.

**RNA transfection**. Transfections with mod-mRNAs (encoding mWasabi, d2eGFP, and reprogramming factors), m-miRNAs, and control siRNAs were performed using Lipofectamin® RNAiMAX™ (RNAiMAX) (Thermo Fisher Scientific). RNA

and RNAiMAX were first diluted in either pH-adjusted Opti-MEM® I Reduced Serum Medium (Opti-MEM) (Thermo Fisher Scientific) or 1× pH-adjusted PBS as indicated in the corresponding figures. pH-adjusted Opti-MEM or pH-adjusted PBS were used as transfection buffers for complex formation between RNAiMAX and RNA. Given that the pH of transfection buffers may affect the resulting transfection efficiency, a range of pH values was tested. The pH of Opti-MEM and 1× PBS (Ambion) was adjusted to indicated values with 1 M NaOH and 1 M HCl at room temperature (RT). For mod-mRNA transfections, 100 ng/μl RNA was diluted 5×, and 5 μl of RNAiMAX per microgram of mod-mRNAs was diluted 10× using either pH-adjusted Opti-MEM or pH-adjusted PBS. After dilution, these components were combined together and incubated for 15 min at RT. For the m-miRNA transfections, a 5 μM (5 pmol/μl) m-miRNA mix was diluted to 0.6 pmol/μl, and 1 μl of RNAiMAX per 6 pmol of m-miRNAs was diluted 10× using either pH-adjusted Opti-MEM or pH-adjusted PBS. The diluted m-miRNA mix and RNAiMAX were mixed together and incubated for

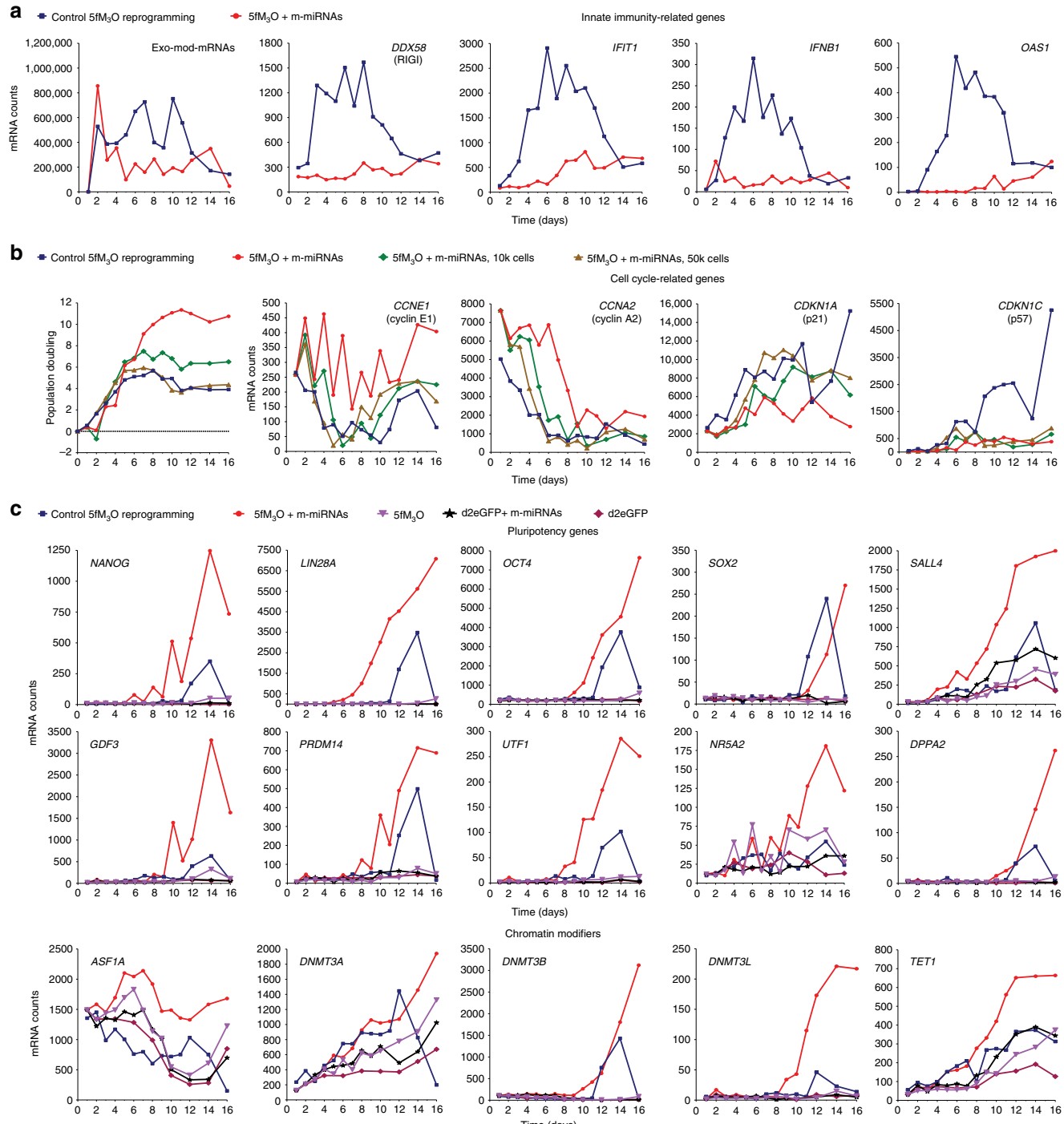

15 min at RT. After incubation at RT, transfection mixtures of mod-RNA mix and/or m-miRNA mix and RNAiMAX were applied to the cell cultures.

**Transfection experiments**. Human primary neonatal fibroblasts were cultured at low confluency in fibroblast medium until transfection. One day before transfection, tissue culture 6-well format dishes (Corning) were coated with Geltrex® Matrix at 100× dilution in plain DMEM/F12 (Thermo Fisher Scientific). Primary patient neonatal and adult fibroblasts were plated onto the Geltrex-coated dishes at 50,000 cells per well of a 6-well format dish. The plating medium was comprised KOSR medium (DMEM/F12 with L-glutamine and no HEPES (catalog # 11320), 20% KOSR, 0.5× MEM non-essential amino acid solution, 0.5× GlutaMAX™ Supplement, 55 μM β-ME, 1× antibiotic–antimycotic solution (all from Thermo Fisher Scientific), and 50 μg/ml L-ascorbic acid) supplemented with 100 ng/ml bFGF, 200 ng/ml B18R (eBioscience), and 5% HI-FBS (Thermo Fisher Scientific). The plated cells were incubated overnight in a low $O_2$ (5%)/5% $CO_2$ tissue culture incubator. The following day, the medium was changed to KOSR medium equilibrated overnight at 5% $O_2$/5% $CO_2$ supplemented with fresh 100 ng/ml bFGF and 200 ng/ml B18R. The volume used per well of a 6-well format dish was 1 ml. RNA transfections were performed as described above. The transfection efficiency of mWasabi mod-mRNA or Alexa Fluor 488-conjugated AllStars Negative siRNA achieved with different transfection buffers was determined by fluorescence-activated cell sorting (FACS) using an LSR II flow cytometer (BD Instruments) and analyzed using FlowJo software (Tree Star).

**Control mRNA reprogramming**. Control feeder-free mRNA reprogramming (Supplementary Fig. 1) was performed as previously described[15]. Specifically, primary neonatal fibroblasts were plated onto dishes coated with CELLstart™ (Thermo Fisher Scientific) in accordance with the manufacturer's instructions at densities ranging from 10,000 to 100,000 cells per well of a 6-well format dish in Pluriton™ medium (Stemgent) supplemented with 200 ng/ml B18R (eBioscience). Media were replaced daily before and after mod-mRNA transfections. B18R supplementation was discontinued the day after the final transfection. Eleven daily 24-h-long transfections of 5fM₃O mod-mRNAs were performed as described above using RNAiMAX and unadjusted Opti-MEM at its original pH of 7.2–7.3 as the transfection buffer. A mRNA dose of 800 ng per well was delivered with each transfection. The amount of mRNA was reduced to 200, 400, and 600 ng on the first three transfections. Adjusting the pH of Opti-MEM in the control mRNA reprogramming, as well as the co-transfection with 20 pmol of m-miRNAs every 48 h, resulting in substantial cytotoxicity and complete cell death by day 4 of reprogramming.

**Optimization of RNA-based reprogramming**. Except for KOSR medium, other media such as E8, N2B27, Pluriton™, and mTeSR™1 (STEMCELL Technologies) failed to support the growth of low-density fibroblast cultures during co-transfections with 5fM₃O mod-mRNAs and m-miRNAs in our initial prescreening experiments. For optimizations and iPSC generation, primary patient's neonatal and adult fibroblasts (except for FD54 and F62) were plated onto Geltrex-coated dishes at densities ranging from 200 to 10,000 cells per well of a 6-well format dish in KOSR medium supplemented with 5% HI-FBS, 100 ng/ml bFGF, and 200 ng/ml B18R. Down syndrome patient's fibroblasts and F62 were reprogrammed using the defined, human recombinant Laminin-521 matrix (Thermo Fisher Scientific), applied according to the manufacturer's instruction. Laminin-521 appeared to be more consistent for reprogramming of fibroblasts than Geltrex. Laminin-521 is also more appropriate for clinical and research applications due to its low batch-to-batch variability. We also tested another defined matrix, CELLstart, and found it to be compatible with the protocol. The plated cells were incubated overnight in a low $O_2$ (5%)/5% $CO_2$ tissue culture incubator. The following day the medium was changed to KOSR medium equilibrated overnight at 5% $O_2$/5% $CO_2$ supplemented with fresh 100 ng/ml bFGF and 200 ng/ml B18R. The first transfection was performed 1 h after changing the medium. RNA transfections were performed as described above using pH-adjusted Opti-MEM or pH-adjusted PBS as indicated in corresponding figures. Differing amounts of mod-mRNAs and m-miRNA were used as described below (for the optimized regimen and single-cell reprogramming) and in corresponding figures. For reprogramming into iPSCs, the mod-mRNA transfection mix was applied first separately followed by the m-miRNA mix. Note that under every transfection condition, the cell growth medium was KOSR medium, whereas complex formation between Lipofectamine and mod-mRNAs was performed in either Opti-MEM or PBS at the indicated pH. Three to seven transfections were performed every 48 h, as shown in Figs. 1b and 2. Regimens with transfections performed every 24, 48, or 72 h were also tested. In the 24 h regimen, 11 consecutive transfections were performed. In the 48 h regimen, up to seven transfections were performed. In the 72 h regimen, a maximum of five transfections were performed. KOSR medium was changed within 20–24 h after each transfection. KOSR medium was equilibrated overnight at 5% $O_2$/5% $CO_2$ before each medium change and supplemented with fresh 100 ng/ml bFGF and 200 ng/ml B18R. The volume used per well of a 6-well format dish was 1 ml. After completing the series of transfections, B18R supplementation was discontinued and KOSR medium supplemented with 100 ng/ml bFGF was changed every day. The cells were grown up to day 18 and then stained with anti-TRA-1-60 antibody (Stemgent, 09-0010) in combination with the horse radish peroxidase (HRP)-conjugated anti-mouse secondary antibody (Thermo Fisher Scientific, 31432) using a standard immunocytochemistry technique. TRA-1-60-positive colonies were visualized with NovaRED HRP substrate (Vector Laboratories). The number of TRA-1-60-positive colonies was counted under Nikon Eclipse TE2000-S inverted microscope with a 10× objective (Supplementary Fig. 18). The reprogramming efficiency was calculated by dividing the number of resulting TRA-1-60-positive colonies by the number of input cells and multiplying by 100%. The representative images of the plates were taken with Sony Alfa A77 Camera equipped with the 50 mm f/2.8 Macro A-Mount Lens. The contrast and brightness of the images were adjusted with Adobe Photoshop. For the optimized RNA-based reprogramming approach (Fig. 1b) performed in a 6-well format dish, the amount of the reprogramming mod-mRNA mix per well per transfection was 600 ng for healthy neonatal fibroblasts and 1,000 ng for adult and disease-associated fibroblasts, and the amount of m-miRNAs per well per transfection was 20 pmol with seven RNA transfections being performed every 48 h using RNAiMAX and Opti-MEM adjusted to pH 8.2 as the transfection buffer. The higher amount of mod-mRNAs (1,000 ng per transfection) appeared to be more consistent for adult and disease-associated fibroblasts. This amount can also be used for neonatal fibroblasts without losing reprogramming efficiency (Fig. 1c, d and Supplementary Fig. 3b, c). The iPSC colonies designated for the maintenance and characterization were manually picked on days 15–18 and plated either on Geltrex-coated tissue culture dishes in E8 medium supplemented with 10 μM Y-27632 or onto a mitomycin C-inactivated human neonatal fibroblast feeder layer in N2B27 medium supplemented with 100 ng/ml bFGF and 10 μM Y-27632. Y-27632 was removed the next day, and the iPSC lines were passaged as described above.

In some experiments, negative siRNA as a control for m-miRNA transfections was used in combination with the reprogramming mod-mRNA mix. Negative siRNA was noted to be highly toxic for cells, reducing cell reprogramming efficiency as compared to reprogramming with mod-mRNAs alone (Fig. 1 and Supplementary Fig. 3).

**Single-cell reprogramming**. To overcome the cell stress caused by FACS, a limiting dilution approach was employed. Human primary neonatal fibroblasts were plated at very low cell densities (<1 cell per well) onto Geltrex-coated 48-well plates in KOSR medium supplemented with 5% HI-FBS, 100 ng/ml bFGF, and 200 ng/ml

**Fig. 7** Reduced innate immunity response and increased cell expansion enhance RNA-based reprogramming. The time-course analysis of gene expression was performed on human primary neonatal fibroblasts (FN2) undergoing reprogramming under different regimens as described in Supplementary Fig. 14. Fibroblasts were subjected to either a control mod-mRNA reprogramming regimen initiated at 50,000 cells per well of a 6-well format dish with mod-mRNA delivered every 24 h as described in Supplementary Fig. 1 (control 5fM₃O reprogramming, blue) or the optimized RNA-based reprogramming protocol as described in Fig. 1b (5fM₃O + m-miRNA), initiated at 500 cells per well (red), 10,000 cells per well (10k, green), or 50,000 cells per well (50k, light brown) of a 6-well format dish with RNA transfections performed every 48 h. As controls, cultures seeded at the initial plating density of 500 cells per well were transfected every 48 h with either reprogramming mod-mRNA alone (5fM₃O, purple) or control d2eGFP mod-mRNA alone (d2eGFP, raspberry), or in combination with reprogramming m-miRNAs (d2eGFP + m-miRNA, black). **a** Graphs showing normalized mRNA counts for exogenous mod-mRNAs (Exo-mod-mRNAs) and innate immunity-related genes throughout the indicated reprogramming regimens as detected by the Nanostring nCounter Gene Expression Assay. **b** Graphs summarizing population doubling (PD) and normalized mRNA counts for a set of cell cycle-associated genes throughout the indicated reprogramming regimens as detected by the Nanostring nCounter Gene Expression Assay. **c** Graphs showing normalized mRNA counts for selected pluripotency genes and chromatin modifier genes throughout the indicated reprogramming regimens as detected by the Nanostring nCounter Gene Expression Assay. The X-axis shows time points (days) at which the samples were collected for analysis during the reprogramming regimens. The Y-axis indicates values of either normalized mRNA counts or PD as specified on the corresponding plots. The analysis of additional genes associated with innate immunity, cell cycle, and pluripotency is shown in Supplementary Fig. 15. The results of the time-course analysis are reproducible using an independent primary neonatal fibroblast line, FN1 (Supplementary Figs. 16 and 17)

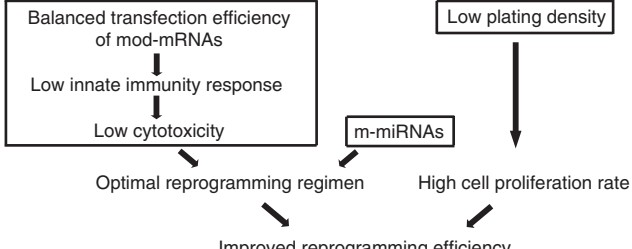

**Fig. 8** The mechanism underlying the enhanced reprogramming activity of the optimized RNA-based approach. The schematic depicts the proposed mechanism by which the optimized RNA-based reprogramming approach improves the efficiency of cellular reprogramming. The reduced cytotoxicity of the optimized mod-mRNA/miRNA-based reprogramming regimen in combination with the low initial cell plating density, which allows for the maximum proliferation of input cells, substantially enhances the efficiency of iPSC generation

B18R as described above. The following day, each well was screened under the microscope to ensure that only 1 cell was seeded per well. The wells with no cells or more than 1 cell were eliminated from the experiment. The amount of RNAs used throughout the regimen was adjusted to the surface area based on 1,000 ng of the reprogramming mod-mRNA mix and 20 pmol of m-miRNAs per transfection per well of a 6-well format dish. For each well of a 48-well format dish, 150 µl of KOSR medium supplemented with bFGF and B18R was used. For each transfection, 100 ng of 5fM$_3$O mod-mRNAs and 2 pmol of m-miRNAs or no m-miRNA mix were used per well of a 48-well format dish. Seven transfections were performed at 48 h intervals as described in Fig. 1b using Opti-MEM adjusted to a pH of 8.2 as the transfection buffer, and KOSR medium was changed within 20–24 h after each transfection. At the end of the transfection series, the medium was changed every day. Cells were grown up to day 18 and then stained with anti-TRA-1-60 antibody. The wells with iPSC colonies were counted. Only wells with more than 10 cells detected at day 18 of the regimen were included for the calculation of reprogramming efficiency. The reprogramming efficiency was calculated by dividing the number of wells with TRA-1-60-positive colonies by the number of wells with surviving input cells and multiplying by 100%. Representative images of the plates were taken with a Sony Alfa A77 Camera equipped with a 50 mm f/2.8 Macro A-Mount Lens. The experiment was performed with three independent human primary neonatal lines.

**Statistical analysis**. For statistical analyses, single pairwise comparisons were calculated using the unpaired two-tailed Student's $t$ test with $P < 0.05$ (*), $P < 0.01$ (**), $P < 0.001$ (***), and $P < 0.0001$ (****) indicating significance. The datasets used for statistical comparison were also analyzed for normal distribution using the Shapiro–Wilk test and for equal variance using the $F$ test. Values are shown as means with error bars depicting the standard deviation (s.d.).

**Immunofluorescence analysis and H&E staining**. Cells were fixed in 4% formaldehyde and permeabilized with 0.2% Triton X (Sigma) in 0.05% saponin solution for 5 min. Cells were blocked with 10% BSA in PBS (Sigma) and 10% donkey or goat normal serum (The Jackson Laboratory) in saponin solution for 1 h and then stained in blocking buffer at 4 °C overnight. Cells were washed with saponin and stained with secondary antibodies for 2 h at RT. Tissues were fixed in 4% paraformaldehyde, embedded in paraffin, sectioned, and stained with hematoxylin and eosin (H&E). For immunofluorescence staining, slides were deparaffinized by submerging in xylene, rehydrated through a gradient ethanol bath, and the antigen was retrieved via boiling in 10 mM citrate buffer, pH 6.0 (Abcam). In the analysis of teratomas, the H&E staining and single-staining immunofluorescence analysis of expression of each germline-specific marker (TUJ1, vimentin, and Endo-A) were performed on separate, consecutively cut sections. Antibodies used were: Alexa Flour 488-conjugated anti-TRA-1-81 (Stemgent, 09-0069; 1:10); Alexa Fluor 555-conjugated anti-SSEA-4 (BD, 560218; 1:50); anti-OCT4 (Santa Cruz Biotechnology, sc-5279; 1:100), anti-NANOG (R&D Systems, AF1997; 1:50); anti-SOX2 (Cell Signaling Technology, 3579S; 1:160); anti-LIN28A (Cell Signaling Technology, 8706S; 1:100); anti-Endo-A (Developmental Studies Hybridoma Bank, TROMA-I; 1:100), anti-β-tubulin III (TUJ1) (Covance, PRB435P; 1:5,000), anti-human vimentin (Abcam, 16700; 1:100). Secondary antibodies used were: Alexa Fluor 594 donkey anti-goat IgG (H + L) (A-11058) 1:250, Alexa Fluor 594 goat anti-mouse IgG (H + L) (A-11005) 1:250, Alexa Fluor 594 goat anti-rat IgG (H + L) (A-11007), Alexa Fluor 594 goat anti-rabbit IgG (H + L) (A-11037), and Alexa Fluor 488 goat anti-rabbit IgG (H + L) (A-11034), all from Thermo Fischer Scientific. After staining, slides were mounted in mounting medium with DAPI (4',6-diamidino-2-phenylindole)

(Vector Laboratories). Images were acquired using a Nikon Eclipse 90i upright microscope using a 10× objective. The contrast and brightness of the images were adjusted with Adobe Photoshop.

**Nanostring nCounter gene expression assay**. Total RNA from the equivalent of 10,000 cells was used for the analysis with the nCounter system, according to the manufacturer's protocol (Nanostring Technologies). nCounter® Gene Expression CodeSets for all experiments were custom-designed by Nanostring Technologies. Supplementary Data 1 includes a CodeSet for the expression analysis of hTERT, selected pluripotency-associated and fibroblast-associated genes described in Supplementary Fig. 10. Supplementary Data 2 includes a CodeSet for innate immunity, cell cycle, and pluripotency genes analyzed in the reprogramming time-course experiment described in Supplementary Fig. 14. The specific exogenous RNA (Exo-mod-mRNA) probe was designed against the 3′-untranslated region (UTR) element from the mouse α-globin gene. Probes for endogenous OCT4, NANOG, SOX2, and LIN28A genes were designed to target specific UTR sequences to prevent cross-reactivity with the reprogramming mod-mRNAs applied to fibroblasts. Digital RNA counting and data analyses were performed by Nanostring Technologies using nSolver® analysis software. Raw RNA counting data were normalized to the mean of the positive control probes for each assay and to the geometric mean of 5 housekeeping genes listed in Supplementary Datas 1 and 2. Probes for housekeeping genes were chosen from the list provided by Nanostring Technologies.

**RNA-Seq analysis**. Indexed samples were pooled, divided, and loaded onto eight lanes of an Illumina HiSeq2000 flow cell yielding 50-bp single-end reads. On average 60 million reads were collected for each sample. The resulting sequences were filtered and trimmed to remove low-quality bases (Phred score < 15) from the 3′ ends of the reads using a custom Perl script. The remaining sequences were mapped to the human genome (hg19) using GSNAP (version 2012-07-20)[25]. After alignment, Cufflinks (version 2.2.1)[26] was used to assemble transcripts and estimate the gene expression values expressed as fragments per kilobase per million (FPKM). In the programming language R, the FPKM values from all 14 samples were used for the principal components analysis.

**In vitro and in vivo differentiation analyses**. The iPSCs were differentiated into cells from a neuronal lineage using a protocol adapted from Hu et al.[27] and Chambers et al [28]. Briefly, iPSC cultures were disaggregated using collagenase type I and feeder-depleted on gelatin for 30 min at 37 °C. The non-adherent cells were collected and plated on a Geltrex-coated dish at a density of 25–30,000 cells/cm$^2$ in N2B27 medium supplemented with 100 ng/ml bFGF and 10 µM Y-27632. Y-27632 was removed the next day, and cells were allowed to expand until 80% confluency. Cells were then disaggregated following the procedure described above using collagenase type I, and iPSC cell aggregates (embryoid bodies) were formed in suspension culture on low-attachment tissue culture plates in N2B27 medium supplemented with 250 ng/ml Noggin (R&D) and 10 µM SB431542 (Stemgent) for the initial 4 days of differentiation. The formed aggregates were induced with neural induction media containing Neurobasal medium supplemented with N2, B27, and heparin for 14 days. The induction phase was followed by the differentiation stage in Neurobasal medium supplemented with N2, B27, 1 µg/ml laminin (Sigma), 100 nM cAMP (Sigma), 200 ng/ml ascorbic acid (Sigma), 10 ng/ml brain-derived neurotrophic factor (Peprotech), 10 ng/ml glial cell-derived neurotrophic factor (Peprotech), and 10 ng/ml IGF-I (Peprotech). At day 84 of differentiation, cells were fixed and processed for immunofluorescence staining with an antibody raised against neuron-specific βIII-Tubulin (TUJ1) as described above.

The iPSCs were differentiated into cells from an endodermal lineage using a previously published protocol[29]. Specifically, iPSCs were feeder-depleted and seeded on a Geltrex-coated dish. These cells were then differentiated as a monolayer at 70% confluency. For the first 4 days, RPMI medium (Gibco) supplemented with 1× glutamine, 450 µM monothioglycerol (MTG) (Gibco), 2 µM CHIR 99021 (Stemgent), 100 ng/ml Activin A (R&D), 50 µg/ml ascorbic acid, 25 ng/ml BMP4 (R&D), 5 ng/ml bFGF, and 10 ng/ml vascular endothelial growth factor (VEGF) (R&D) was used. On day 5, cells were disaggregated with Accutase and seeded at 250,000 cells per well on a Geltrex-coated dish. Cells were fed every 2 days until day 18 with media supplemented with cytokines provided above. At day 18 of differentiation, cells were immunostained with an antibody against endoderm-specific cytokeratin Endo-A using an immunostaining protocol as described above.

The iPSCs were differentiated into cells of a cardiomyocyte lineage using a previously published protocol with modifications[30]. Briefly, cells were feeder-depleted and seeded on a Geltrex-covered dish. These cells were then dissociated using collagenase type I and placed in a suspension culture for embryoid body formation in basic medium containing StemPro34 (Invitrogen), 2 mM glutamine, 150 µg/ml Transferrin (Roche), 400 µM MTG, 50 µg/ml ascorbic acid, and 10 ng/ml BMP4. The following day, the cell aggregates were switched to induction medium containing StemPro34 (Invitrogen), 2 mM glutamine, 150 µg/ml transferrin (Roche), 400 µM MTG, and 50 µg/ml ascorbic acid, and supplemented with the following cytokines: days 1 and 2: activin A (6 ng/ml), BMP4 (10 ng/ml) (R&D),

and bFGF (2.5 ng/ml); days 3–5: VEGF (1.25 ng/ml) (R&D) and DKK1 (150 ng/ml) (R&D); days 8–14: VEGF (1.25 ng/ml), DKK1 (150 ng/ml), and bFGF (2.5 ng/ml). Beating cardiomyocytes were observed at day 14. The videos were taken using the 3i Marianas spinning disk confocal (3i-Intelligent Imaging Innovations Inc., Denver CO, USA) in a bright-field mode via a transmitted light path. The software used was Slidebook 5.5 (3i-Intelligent Imaging Innovations).

For teratoma analysis, iPSCs ($1 \times 10^6$ cells) were injected subcutaneously into the flank of 6-week-old female SCID mice (The Jackson Laboratory). For each iPSC line, two mice were injected. The animals were monitored for 2 months for the formation of a tumor with terminal size limited to 2 cm. If both animals injected with a single iPSC line developed tumors, only one randomly selected tumor was analyzed. No other randomizations were performed. Investigators were not blinded to the experimental groups. All experiments with mice were performed in accordance with the regulations and approval of the University of Colorado Denver Institutional Animal Care and Use Committee.

**Karyotyping.** The cytogenetic analysis was performed by the University of Colorado Cancer Center Molecular Pathology Shared Resource on fixed cell pellets using standard GTL banding (G-banding) of metaphase chromosomes. Metaphase spreads were captured using a CCD camera and karyotyping was carried out using BandView software (Applied Spectral Imaging Inc). Chromosome classification followed ISCN (2013) guidelines. From 13 to 15 chromosome spreads were analyzed from each cell line.

**Cell line identity confirmation.** The identity of iPSC lines derived from FN2, F50, and F50S was confirmed by STR genotyping using the STR genotyping AmpFLSTR® *Identifiler*® PCR Amplification Kit (Thermo Fischer Scientific). The STR genotypes of the generated iPSC lines matched those of primary fibroblast lines from which they had been derived. The identity of disease-associated iPSC lines was confirmed by the presence of either corresponding disease-causing mutations or trisomy 21 (for Down syndrome lines).

**Bisulfite sequencing.** The promoter region of the human *OCT4* gene was analyzed as previously described with minor modifications[21]. Specifically, genomic DNA from FN2 fibroblasts, FN2-derived iPSCs (IN2–5) and hESCs (H9) was treated with EpiTect Bisulfite Kit (Qiagen) according to the manufacture's instruction. The human *OCT4* promoter sequence was amplified by PCR using primers: bi OCT4-S, 5′-TAGTTGGGGATGTGTAGAGTTTGAGA-3′ and bi OCT4-AS, 5′-TAAAC-CAAAACAATCCTTCTACTCC-3′. Generated PCR products were subcloned into pGEM-T Easy vector system (Promega). Finally, ten random clones corresponding to each original cell line were picked and sequenced.

**Immunoblot analysis.** Antibodies used for standard immunoblotting were: anti-p21 (Santa Cruz Biotechnology, sc-469; 1:250), anti- β-actin (Santa Cruz Biotechnology, sc-1616; 1:500), anti-rabbit IgG-HRP (Santa Cruz Biotechnology, sc-2004; 1:5,000), anti-goat IgG-HRP (Santa Cruz Biotechnology, sc-2020; 1:5,000). Signals were detected with "chemiluminescent" (Thermo Fisher Scientific). After the detection of p21, the same blot was stripped and reprobed with the anti-β-actin antibody as a loading control (see Supplementary Fig. 7b for uncropped scans of the blot).

**Telomere length analysis.** The length of telomeres was measured using quantitative PCR (qPCR)[31]. Specifically, DNA was isolated from low passage (F50) and senescent (F50S) parental fibroblasts, iPSC lines derived from F50 (I50-2 and I50-3) and F50S (I50S-1 and I50S-2), and human ESCs (H1). qPCR analysis was performed with telomere-specific and human single copy gene (SCG) control primers, using LightCycle 480 (Roche). Telomere-specific primers are: teloF, 5′-CGGTTTGTTTGGGTTTGGGTTTGGGTTTGGGTTTGGGTT-3′ and teloR, 5′-GGCTTGCCTTACCCTTACCCTTACCCTTACCCTTACCCT-3′. Human SCG control primers are: 36B4F, 5′-CAGCAAGTGGGAAGGTGTAATCC-3′ and 36B4R, 5′-CCCATTCTATCATCAACGGGTACAA-3′.

**Telomerase activity analysis.** Telomerase activity was measured with the TRA-PEZE Telomerase Detection Kit (Chemicon) according to the manufacturer's instructions. CHAPS (1×) lysis buffer was used to obtain extracts from parental fibroblasts (F50), iPSC lines derived from F50 (I50-2 and I50-3), and iPSC lines derived from senescent F50S line (I50S-1 and I50S-2) and H1 ESC line. About 2,000 cells were assayed for each telomeric repeat amplification protocol assay, and 800 cell equivalents were loaded into each well of a 15% non-denaturing TBE (Tris borate, EDTA) polyacrylamide gel. Each sample was heat inactivated for 15 min at 85 °C to assess the background of the assay. Data were analyzed with Fluoro Chem HD2 scanner.

**Data availability.** The authors declare that all data supporting the findings of this study are available within the article and its supplementary information files or from corresponding author upon reasonable request. Nucleotide sequences for RNA-Seq data have been deposited in the Gene Expression Omnibus database at NCBI under accession code GSE97265.

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

## Acknowledgements

We are grateful for funding support from the National Institutes of Health (R01AR059947 and T32AR007411-33), the US Department of Defense (W81XWH-12-1-0606), the Foundation for Ichthyosis and Related Skin Types (F.I.R.S.T.), Dystrophic Epidermolysis Bullosa Research Association (DEBRA) International, The King Baudouin Foundation's Vlinderkindje Fund, the Epidermolysis Bullosa (EB) Research Partnership, the EB Medical Research Foundation, the SOHANA Research Fund, the Linda Crnic Institute for Down Syndrome, and the Gates Frontiers Fund. We also thank Skin Morphology and Phenotyping Core and Flow Cytometry Core supported by the University of Colorado (UC) Skin Diseases Research Core Center (P30AR057212); Colorado's NIH/NCI Cancer Center Molecular Pathology Shared Resource (P30CA046934), the UC Advanced Light Microscopy Core supported in part by NIH/NCATS Colorado CTSI Grant (UL1 TR001082). Sequencing and bioinformatic analysis were supported in part by the Biostatistics/Bioinformatics and Genomics and Microarray Shared Resources of Colorado's NIH/NCI Cancer Center (P30CA046934). We thank Dr. Christopher Korch for STR genotyping of cell lines.

## Author contributions

I.K., D.R.R. and G.B. conceived the experiments, interpreted results, and wrote the manuscript. I.K. and G.B. performed all optimizations of reprogramming conditions, conducted all reprogramming experiments, analyzed the data, and generated figures. S. M.M., M.P., X.C. and A.J. assisted in analyzing iPSCs and optimizing RNA transfection regimens. D.P.A. and K.L.J. performed bioinformatics analysis of expression data. A.G. and J.C.C. assisted with flow cytometry analysis. A.M.G.P. and M.F.J. assisted in preparation of patient's fibroblasts.

## Additional information

**Competing interests:** I.K., D.R.R., and G.B. are co-inventors on a patent application entitled Methods and Compositions for Reprogramming Cells; PCT Application No. PCT/US2016/063258. The remaining authors declare no competing financial interests.

