## [Peer Review File · Nature Communications]

Reviewers' comments:

Reviewer #1 (Remarks to the Author):

In this manuscript, the authors developed an optimized RNA-based reprogramming approach which combined the six reprogramming factors (OCT4/SOX2/KLF4/cMYc/LIN28/NANOG) with miRNAs-367/302s. Tra-1-60 positive colonies can be induced from the multiple normal and disease-specific human fibroblast cell lines with a high efficiency. Up to 90% of individually plated human primary neonatal fibroblasts can be reprogramed using this protocol. In addition, to analyze the synergistic effects of the mod-mRNA and m-miRNAs in iPS reprogramming, the expression pattern of the innate immunity and pluripotency related genes were examined.

The approach reported here showed a high reprogramming efficiency. However, the characteristics of the iPS cell lines established by this protocol should be further analyzed and compared with that of human ESCs. In addition, several concerns below should be addressed to make the conclusions more solid.

1. In figure 3, the data of the characteristics of the iPS cell lines are preliminary. The authors should further analyze the identity of the iPSCs established by the new protocol, including 1) detecting the expression of pluripotent genes at mRNA levels; 2) comparing the global gene expression profiles of iPSCs with that of human ESCs; 3) Confirming the expression of more pluripotent genes at protein levels such as SOX2, LIN28, UTF1 and REX1; 4) testing the mesodermal differentiation potential of iPSCs in vitro
2. In figure 1a, S2a and S3a, the transfection efficiency of the RNAs in opti-MEM-8.6 was higher than that in Opti-MEM-7.8. However, in figure 1C and S2b, more tra-1-60 positive colonies were induced in Opti-MEM-7.8 compared to that induced in opti-MEM-8.6. Any explanations?
3. To assess the robustness of the protocol on older patients, the author induced iPSCs from the senescent fibroblasts that were generated by serially passaged in vitro. However, the senescent fibroblasts induced in vitro are not identical to the fibroblasts from the older patients. It would be more convincing to reprogram the fibroblasts isolated from the older patients directly.
4. In figure 4, the biologic repetition should be provided at each time point during the reprogramming process and the controls of iPSCs and hESCs should also be provided.
5. Scale bars should be annotated in figure legend 4.

Reviewer #2 (Remarks to the Author):

This manuscript from Kogut et al addresses several optimizations in RNA-based reprogramming methods, including the synergistic combination of modified mRNAs (mod-mRNAs) and miR 302/367 cluster mimics, substitution of M3O OCT4 for wildtype mod-mRNA in the 6-factor cocktail, buffer pH, fibroblast density, the amount of RNA transfected, and the frequency of transfections. Taken together, these methods result in a significant improvement in reprogramming efficiencies. The authors also report reprogramming efficiencies using their optimized RNA-based method in several individual neonatal and adult fibroblast lines, senescent fibroblasts, and 4 disease patient-derived fibroblast lines.

Although this is a straightforward report that could be of interest to iPSC researchers, especially to those who are using the modified mRNA-based reprogramming, it would be important to show that similar optimizations (number of factor cocktail, buffer pH, fibroblast density, the amount of vectors transfected) could improve significantly the other non-integrating ways of reprogramming.

Nevertheless the study remains highly incremental. Thus, this reviewer is uncertain if NCOMM is the right journal to publish the author's findings.

We thank the reviewers for their thorough review of our manuscript for their suggestions and comments for improvement. Our detailed responses to the reviewer's comments/suggestions are highlighted in bold.

Reviewer #1 (Remarks to the Author):

In this manuscript, the authors developed an optimized RNA-based reprogramming approach which combined the six reprogramming factors (OCT4/SOX2/KLF4/cMYC/LIN28/NANOG) with miRNAs-367/302s. Tra-1-60 positive colonies can be induced from the multiple normal and disease-specific human fibroblast cell lines with a high efficiency. Up to 90% of individually plated human primary neonatal fibroblasts can be reprogrammed using this protocol. In addition, to analyze the synergistic effects of the mod-mRNA and m-miRNAs in iPS reprogramming, the expression pattern of the innate immunity and pluripotency related genes were examined

The approach reported here showed a high reprogramming efficiency. However, the characteristics of the iPS cell lines established by this protocol should be further analyzed and compared with that of human ESCs. In addition, several concerns below should be addressed to make the conclusions more solid.

1. In figure 3, the data of the characteristics of the iPS cell lines are preliminary. The authors should further analyze the identity of the iPSCs established by the new protocol, including 1) detecting the expression of pluripotent genes at mRNA levels; 2) comparing the global gene expression profiles of iPSCs with that of human ESCs; 3) Confirming the expression of more pluripotent genes at protein levels such as SOX2, LIN28, UTF1 and REX1; 4) testing the mesodermal differentiation potential of iPSCs in vitro.

- 1) We apologize that we did not emphasize the extensive characterization that we performed with the generated induced pluripotent stem cell (iPSC) lines in the text. Many of the requested characterizations were performed but not properly discussed in the original manuscript. We have now created a section in the text entitled “Molecular and functional characterization of the generated iPSC lines” that describes the type of characterization we performed (page 8 in the revised manuscript). The main Figure 3 does not include all of our data due to space constraints. Multiple supplementary files (Supplementary Figures 9-14; Supplementary Videos 1-5; Supplementary Tables S1 and S2) include additional data characterizing our iPSC lines.**
- 2) The reviewer suggested that we examine the expression of pluripotency genes at the mRNA level and compare the global gene expression profiles of iPSC lines with that of human Embryonic Stem Cells (hESCs). Fig. 3a shows a principal component analysis (PCA) of global mRNA transcription profiles captured by RNA sequencing (RNA-Seq). The analysis includes parental fibroblasts, iPSCs and human ESCs labeled as H1 and H9. Our iPSC lines and hESC lines cluster together as expected. We apologize if this comparison was not clear in our original submission. We have revised the figure and highlighted the data for the H1 and H9 hESC lines in a different color to make them more visible. We also modified the text accordingly (page 8, lines 11-13). We have also performed additional mRNA analysis (the Nanostring nCounter Gene Expression Assay) to further confirm the expression of the pluripotency genes at the mRNA level as was requested. These data are summarized in the Supplementary Fig. 11**
- 3) In the original manuscript, we confirmed the expression of three pluripotency markers (Oct4, Nanog, and TRA-1-81) at the protein level in 16 different iPSC clones. In addition, all our reprogrammed iPSC colonies were visualized by TRA-1-60 staining, making it the forth pluripotency marker analyzed at the protein level. In the revised manuscript, we**

have added the analysis of three additional pluripotency markers (SOX2, LIN28A, and SSEA-4) at the protein level as was requested, and these data are now in Fig. 3b and Supplementary Figs 9e, 12 and 13. In addition to mRNA and protein expression analyses, we also performed more stringent pluripotency tests on our iPSC lines. These tests include teratoma formation *in vivo* (Fig. 3e, Supplementary Figs. 9f and 14) and *in vitro* differentiation into cell types of all three germ layers (Fig. 3d, Supplementary Fig. 9g, and Supplementary Videos 1-5). These assays of pluripotency are more informative since they confirm the functionality and differentiation potential of our iPSC lines.

- 4) The reviewer suggested that we test the mesodermal differentiation potential of iPSCs *in vitro*. The *in vitro* differentiation of our iPSC lines into mesoderm was confirmed by deriving beating cardiomyocytes in the original manuscript. We now more clearly emphasized this in the text (page 8, line 19-20). We also added 3 additional movies with beating cardiomyocytes derived from several iPSC lines (see Supplementary Videos 1-5). In addition, all of the iPSC lines reported in this manuscript were successfully differentiated into a mesodermal lineage in teratoma induction experiments (see Fig. 3e, Supplementary Figs. 9f and 14), confirming the mesodermal differentiation potential of our iPSCs.

2. In figure 1a, S2a and S3a, the transfection efficiency of the RNAs in opti-MEM-8.6 was higher than that in Opti-MEM-7.8. However, in figure 1C and S2b, more *tra-1-60* positive colonies were induced in Opti-MEM-7.8 compared to that induced in opti-MEM-8.6. Any explanations?

We apologize that we did not discuss this observation in the text. The regimen performed with Opti-MEM at pH 8.6 is cytotoxic, probably due to the degradation of mRNA at this higher pH. Degraded RNA will most likely increase the innate immunity response, which in turn will induce cytotoxicity and reduce the reprogramming efficiency. We now included this explanation in the revised manuscript (page 5, line 21-26).

3. To assess the robustness of the protocol on older patients, the author induced iPSCs from the senescent fibroblasts that were generated by serially passaged *in vitro*. However, the senescent fibroblasts induced *in vitro* are not identical to the fibroblasts from the older patients. It would be more convincing to reprogram the fibroblasts isolated from the older patients directly.

We agree with the reviewer that senescent fibroblasts are not identical to fibroblasts from older patients. As requested, we successfully reprogrammed fibroblasts from a 62-year-old patient with an efficiency that surpassed the efficiency of reprogramming of fibroblasts from a 50-year-old individual and included these results in the manuscript (see Table 1 and Supplementary Fig. 8). However, our ability to reprogram senescent fibroblasts is a much more stringent test of the robustness of our reprogramming protocol. We would like to refer the reviewer to the paper published by Lapasset et al (2011). These authors assessed the ability of lentiviruses encoding either a standard 4-factor cocktail (Oct4, Sox2, Klf4, cMyc) or a 6-factor cocktail (4 standard factors +Nanog and Lin28A), similar to the cocktail of modified mRNA that we use, to reprogram fibroblasts from the elderly. They used fibroblasts isolated from a 74-yr-old donor. These fibroblasts were still able to undergo a substantial number of population doublings before they became completely senescent. When a 4-factor cocktail was used for reprogramming, the authors succeeded in reprogramming the proliferative fibroblasts but not the matching senescent fibroblasts. Only when a 6-factor cocktail was used, were the authors able to

reprogram both parental and senescent fibroblasts with similar efficiencies. Similar results were obtained when a matching pair of proliferative and replication senescent embryonic fibroblasts was used. Therefore, these results confirm that replicative senescence is a stronger barrier to reprogramming than the age of the patient. For this reason, we reprogrammed replication senescent fibroblasts as a more stringent test of the robustness of our reprogramming protocol. We modified the text and included senescence as a more stringent test of the protocol's robustness (page 7, line 25-26).

4. In figure 4, the biologic repetition should be provided at each time point during the reprogramming process and the controls of iPSCs and hESCs should also be provided.

The data in Fig. 4 were replicated on an independent fibroblast line (FN1) and these data were shown in Supplementary Figures in the original manuscript. We apologize for not stating this clearly in the text and figure legend. We have clarified this in the revised manuscript (see page 8-9, section: "The optimized RNA-based approach reduces the expression of innate immunity genes and leads to the robust activation of pluripotency-associated genes", Fig. 4, and Supplementary Figs. 16-18).

As to the addition of iPSC and ESC controls, we are not sure how these controls are relevant. The experiment shows the response of fibroblasts to reprogramming RNA transfections during the time course of reprogramming, 16 days, with the ultimate goal of generating iPSCs. The best control for baseline activation prior to transfection is fibroblasts on Day 1 that were collected and analyzed before the first transfection, and those data are shown. The use of established iPSCs and ESCs as controls for levels of expression of pluripotency genes at the end of reprogramming is not a fair comparison, since at the end of reprogramming (before colonies are picked) the culture consists of a mixture of fully reprogrammed, partly reprogrammed and non-reprogrammed cells. In addition, different cell culture media are used for reprogramming vs maintenance of iPSCs or hESCs. All of these differences guarantee that the level of pluripotency gene expression in iPSCs and ESCs will be significantly different from the level observed at the end of reprogramming.

5. Scale bars should be annotated in figure legend 4.

The time course experiment was performed using the Nanostring nCounter Gene expression assay. To avoid confusing these data with RT-PCR results, we have modified the figures and labeled the Y axis as mRNA counts (instead of relative expression). The values on the Y scale are Nanostring mRNA counts normalized to the mean of the positive control probes for each assay and to the geometric mean of housekeeping genes. We have also annotated the figure legend accordingly. Plotting normalized mRNA counts is the standard way of presenting Nanostring data.

Reviewer #2 (Remarks to the Author):

This manuscript from Kogut et al addresses several optimizations in RNA-based reprogramming methods, including the synergistic combination of modified mRNAs (mod-mRNAs) and miR 302/367 cluster mimics, substitution of M3O OCT4 for wildtype mod-mRNA in the 6-factor cocktail, buffer pH, fibroblast density, the amount of RNA transfected, and the frequency of transfections. Taken together, these methods result in a significant improvement in reprogramming efficiencies. The authors also report reprogramming efficiencies using their optimized RNA-based method in several individual neonatal and adult fibroblast lines, senescent fibroblasts,

and 4 disease patient-derived fibroblast lines.

Although this is a straightforward report that could be of interest to iPSC researchers, especially to those who are using the modified mRNA-based reprogramming, it would be important to show that similar optimizations (number of factor cocktail, buffer pH, fibroblast density, the amount of vectors transfected) could improve significantly the other non-integrating ways of reprogramming.

Nevertheless the study remains highly incremental. Thus, this reviewer is uncertain if NCOMM is the right journal to publish the author's findings.

We respectfully disagree with this reviewer that our study is incremental. We would like to emphasize that our RNA-based reprogramming approach will be very valuable for both clinical and basic research applications due to its extremely high efficiency and safety.

First, it may allow researchers, including us, to get FDA approval for the first iPSC-based clinical trial in the US. The low efficiency of reprogramming protocols increases the chance of accumulation of deleterious mutations in iPSCs, hampering the potential clinical application of this technology. In fact, the first clinical trial for the use of iPSCs for treatment of macular degeneration, that was initiated in Japan, was suspended due to the presence of a cancer-associated mutation in one patient's iPSCs (Garber, 2015). The iPSCs used in this trial were produced using a DNA-based episomal plasmid approach, which rarely exceeds a reprogramming efficiency of 0.035% (Rao et al, 2012). Therefore, we believe that the FDA would be more likely to approve the use of our reprogramming approach for generating iPSCs for a clinical trial than episomal- or Sendai virus-based reprogramming approaches.

Second, our reprogramming method addresses the current limitations of reprogramming techniques used for studying the mechanisms of induced pluripotency in human somatic cells. One of the most common strategies for addressing reprogramming mechanisms relies on using genetically modified cells harboring a doxycycline-inducible cassette encoding the reprogramming factors. We believe that, due to its high efficiency, our non-integrating reprogramming method can also offer a unique approach for elucidating the mechanisms of reprogramming of genetically unmodified primary somatic cells, which will be of interest to many researchers in the reprogramming field, not just those who are primarily interested in RNA-based reprogramming.

Finally, there is no reason to believe that the methods we have optimized for the transfection of modified mRNA and miRNAs into fibroblasts would have an impact on the delivery of other non-integrating ways of reprogramming such as episomal- or Sendai virus-based reprogramming approaches. In fact, there are several limitations of the episomal and Sendai virus methods that make it technically impossible to apply our combined optimized methods to these approaches, and we have briefly summarized these limitations below:

- 1. We discovered that it was critical to modulate the stoichiometry of reprogramming factors to achieve a high level of efficiency. It is relatively easy to control the precise levels of expression of different reprogramming factors when using mRNA, but almost impossible when using episomes or Sendai virus vectors.**
- 2. It takes a substantial amount of passaging to eliminate residual episomes or Sendai virus vector particles from already established iPSCs. Dr. Yamanaka showed in his original**

studies that exogenous retroviruses used to deliver reprogramming factors had to be silenced at a precise moment of reprogramming since residual expression of exogenous factors triggers spontaneous differentiation of iPSCs. Because the half-life of mRNA is only 12-24 hrs as indicated in Warren et al (2010), we do not face the problem of residual expression of exogenous factors post-reprogramming, which helps to increase reprogramming efficiency and stability of iPSCs.

- 3. The buffer we used was optimized to improve the transfection efficiency of modified mRNAs. It does not work with episomal DNA, which in fact requires nucleofection to be delivered efficiently into fibroblasts.**
- 4. Initiating reprogramming at a reduced cell density is critical for improving reprogramming efficiency, since there is a direct correlation between rate of cell division and efficiency of reprogramming. However, the toxicity of Sendai virus vectors and nucleofection used for delivering episomes makes it impossible to reduce cell density with these approaches. In fact, we show in our manuscript that even the conventional mRNA-based reprogramming method, which relies on everyday transfection of a high amount of mRNA, requires the initial use of high density cultures due to the high cytotoxicity of this regimen (Supplementary Fig. 1).**

REVIEWERS' COMMENTS:

Reviewer #1 (Remarks to the Author):

In the revised version, the authors have provided more data to demonstrate the identity of the iPSCs established by their new protocol. The authors also presented the results more clearly. Most of my concerns have been solved. However, one important question should be addressed as follows:

The author should explain why the pictures in figure 3e-IN2-1 (mesoderm) and figure 3e-IN2-1 (endoderm) are the same? If these pictures are immunofluorescence of double antibody staining (Vimentin and Endo A), they should be shown as a red-fluorescent vision (Alexa Fluor 594) and a green-fluorescent vision (Alexa Fluor 488). However, these two pictures are both shown as a red-fluorescent vision.

And similar question to figure S9f-150S-1(mesoderm) and figure S9f-150S-1(endoderm)

We thank the reviewers for their thorough review of our manuscript. Our detailed response to the reviewer's comment is highlighted in bold.

Reviewer #1 (Remarks to the Author):

In the revised version, the authors have provided more data to demonstrate the identity of the iPSCs established by their new protocol. The authors also presented the results more clearly. Most of my concerns have been solved. However, one important question should be addressed as follows:

The author should explain why the pictures in figure 3e-IN2-1 (mesoderm) and figure 3e-IN2-1 (endoderm) are the same? If these pictures are immunofluorescence of double antibody staining (Vimentin and Endo A), they should be shown as a red-fluorescent vision (Alexa Fluor 594) and a green-fluorescent vision (Alexa Fluor 488). However, these two pictures are both shown as a red-fluorescent vision.

And similar question to figure S9f-150S-1(mesoderm) and figure S9f-150S-1(endoderm)

The images presented in Fig. 3e (IN2-1-mesoderm and IN2-1-endoderm), as well as in Fig.S9f (I50S-1-mesoderm and I50S-1-endoderm), are the same fields of view in consecutive teratoma sections, which happen to contain both mesodermal and endodermal lineages. The single-staining immunofluorescence for Endo-A and Vimentin was performed separately on consecutive sections using Alexa Fluor 594 –conjugated secondary antibodies. No double antibody/antigen staining was performed on any of the sections to avoid high background. H&E images are also part of the same consecutive section series. We now recognize that we did not clearly indicate that the immunostaining was performed on consecutive sections. We have now modified the Methods Section and included the following statement into the “Immunofluorescence analysis and H&E staining” subsection:

“In the analysis of teratomas, the H&E staining and single-staining immunofluorescence analysis of expression of each germ line specific marker (TUJ1, vimentin, and Endo-A) were performed on separate, consecutively cut sections.”

We also clarified that consecutive sections were used for analysis in the corresponding figure legends (Fig. 5e and 6e; Supplementary Fig. 13).